# Single-cell analyses reveal transient retinal progenitor cells in the ciliary margin of developing human retina

Birthe Dorgau[1,5], Joseph Collin[1,5], Agata Rozanska[1], Darin Zerti[1,2], Adrienne Unsworth [1], Moira Crosier[1], Rafiqul Hussain[1], Jonathan Coxhead[1], Tamil Dhanaseelan[1], Aara Patel[3], Jane C. Sowden[3], David R. FitzPatrick [4], Rachel Queen[1] ✉ & Majlinda Lako [1] ✉

The emergence of retinal progenitor cells and differentiation to various retinal cell types represent fundamental processes during retinal development. Herein, we provide a comprehensive single cell characterisation of transcriptional and chromatin accessibility changes that underline retinal progenitor cell specification and differentiation over the course of human retinal development up to midgestation. Our lineage trajectory data demonstrate the presence of early retinal progenitors, which transit to late, and further to transient neurogenic progenitors, that give rise to all the retinal neurons. Combining single cell RNA-Seq with spatial transcriptomics of early eye samples, we demonstrate the transient presence of early retinal progenitors in the ciliary margin zone with decreasing occurrence from 8 post-conception week of human development. In retinal progenitor cells, we identified a significant enrichment for transcriptional enhanced associate domain transcription factor binding motifs, which when inhibited led to loss of cycling progenitors and retinal identity in pluripotent stem cell derived organoids.

Retina is the innermost, light-sensitive tissue that lines the back of the eye and is vital for light sensing and image processing. The retina is derived from a germinal zone in the optic vesicle in which neuroepithelial cells proliferate to give rise to the six types of retinal neuronal and one glial cell type, organized within three different layers. All these cell types derive from retinal progenitor cells (RPCs) in an orderly spatio-temporal manner that has been well studied in vertebrates[1]. Great progress has been achieved in the last 5 years providing both gene expression and epigenetic profiles of the developing human retina[2–4]. Single cell (sc) RNA-Seq studies have enabled molecular characterisation of all the retinal cell types and have provided the developmental trajectories that lead to retinal cell specification from RPCs during retinal development[5–9]. The transcriptomic studies have

been complemented with sc assays for transposase-accessible chromatin sequencing (scATAC-seq), enabling identification of key transcription factors relevant to specific retinal cell fates and the gene regulatory networks (GRNs) that underline the cell-state changes[7,10–12]. Notwithstanding, these studies have not provided as yet any information on the spatial resolution of RPCs or retinal neurons.

The retina of fish and amphibians continuously integrates new neurons generated from RPCs residing in the distal tip of the retina known as the ciliary margin zone (CMZ). In chicks the CMZ also contributes to retinal neurogenesis during development, but the adult chick CMZ cells have a more restricted potential as they only contribute to a small fraction of the retina, and moreover do not participate in retinal regeneration following injury[13]. In mice, the ciliary body

[1]Biosciences Institute, Newcastle University, Newcastle, UK. [2]Department of Biotechnological and Applied Clinical Sciences, University of L'Aquila, L'Aquila, Italy. [3]UCL Great Ormond Street Institute of Child Health and NIHR Great Ormond Street Hospital Biomedical Research Centre, University College London, London, UK. [4]MRC Human Genetics Unit, Institute of Genetics and Cancer, University of Edinburgh, Edinburgh, UK. [5]These authors contributed equally: Birthe Dorgau, Joseph Collin. ✉e-mail: rachel.queen@ncl.ac.uk; majlinda.lako@ncl.ac.uk

which arises from the CMZ was postulated to contain a population of pigmented cells that were able to form clonogenic spheres and could be differentiated to express marker genes found in photoreceptors, bipolar and Muller glia cells[14]. However subsequent studies demonstrated that these pigmented ciliary epithelial cells fail to differentiate into retinal neurones in vitro or in vivo[15]. Instead, progenitors distinct from the classical RPCs, characterised by Msx1 expression and located in the proximal CMZ, were shown to give rise to non-pigmented ciliary epithelial cells and multipotent neural RPCs[16]. Whether the human CMZ displays comparable properties during retinal development remains unclear.

Herein we generated a scRNA-Seq atlas of 24 samples spanning 13 time points from 7.5–21 PCW of human development, enabling identification of early and late RPCs and their transition to neurogenic progenitors and retinal neurons. Complementing scRNA-Seq with spatial transcriptomics (ST) we demonstrate the transient presence of early RPCs in the CMZ of developing human retina. Complementary scATAC-Seq identified key transcription factors and signalling pathways that underline RPCs proliferation and differentiation. Together, these comprehensive single cell analyses shed light on the molecular mechanisms governing human retinal development and provide guidance on the generation of RPCs and differentiated retinal cell types from human pluripotent stem cells (PSCs).

## Results

### scRNA-Seq atlas of human developing eyes and retinas

To better understand the emergence of RPCs, their heterogeneity and differentiation to retinal cell types, we performed scRNA-Seq of 11 embryonic and foetal eyes spanning 7.5–15 PCW and 13 retinal samples from 10–21 PCW (Supplementary Data 1). Each sample was embedded using Uniform Manifold Approximation and Projection (UMAP) and clustered using Seurat graph-based clustering (Fig. S1, Supplementary Data 1). In our earlier study of human retinal development[2], we reported expression of RPC markers (e.g. VSX2) at the peripheral margin of developing human retina as early as 6.5 PCW, and at the outer neuroblastic zone of central retina at 7.8 PCW, marking this time period an important developmental window of human RPC divergence. To better understand this process at the single cell level, we first focused our scRNA-Seq analysis in the 7.5–8.5 PCW embryonic eyes, revealing the presence of retinal and non-retinal cell clusters (Fig. S1A). At this stage of human development, the neuroepithelium-derived optic vesicle is surrounded by periocular mesenchyme (POM), which is derived from neural crest and mesoderm, and contributes to anterior and non-neural ocular tissue development. In accordance, we identified neural crest cell clusters adjacent to POM and corneal endothelial and corneal stromal cells[17] (Fig. S1A). The POM cell clusters of 7.5–8.5 PCW eyes displayed high expression of characteristic markers described in other species such as collagen chains (COL3A1, COL5A1, COL1A1), proteoglycan decorin (DCN)[18] and latent Transforming Growth Factor Beta Binding Protein 1 (LTBP1)[19]. Other non-retinal cell clusters including extraocular muscle, ocular surface epithelium, red blood cells, monocytes and macrophages were also identified in most eye samples of this developmental window (Fig. S1A, Supplementary Data 1).

Amongst the retinal cell clusters, RPCs and retinal ganglion cells (RGCs) were identified in all 7.5–8.5 PCW eyes, consistent with our previous findings of RGC presence at the basal side of the inner neuroblastic zone at 8 PCW[2]. A recent scRNA-Seq study has documented the presence of early and late RPCs and their transcriptional signature[5]. In accordance, we found that RPCs clusters of 7.5 and 7.7 PCW eyes displayed high expression of characteristic early RPC markers such as SFRP2, RAX, PAX6, VSX2, ZIC2, ZIC1, SIX3, SIX6[20] (Supplementary Data 1). While the expression of these markers was maintained in RPCs found in the 8 and 8.5 PCW eye specimens, expression of late RPC markers such as CCND1, ASCL1 and HES6[5] became prominent, suggesting the

co-existence of early and late RPCs. A proliferating cone photoreceptor cluster marked by high expression of proliferation (TOP2A, PCNA), neurogenic progenitors (HES6) and cone markers (RXRG, GNB3) was detected in two eye specimens obtained at 7.7 and 8 PCW (Fig. S1A) as well as one of the 10 PCW samples (Fig. S1B). In the latter sample, the proliferating cone cluster was distinct to the mature cones. Notably a small cluster of horizontal cells was detected in two specimens of 8 and 8.5 PCW of development (Fig. S1A), corroborating the first reported emergence of immature horizontal cells at day 59 of human development[4]. A retinal pigment epithelium (RPE) cell cluster with high expression of characteristics markers namely PMEL, TYRP1, MITF and GJA1 was identified in all eyes of this stage (Fig. S1A, Supplementary Data 1).

A recent scRNA-Seq study provided evidence of the existence of neurogenic RPCs in addition to the early and late RPCs, based on gene expression profiling[5]. Sridhar and colleagues[6] further defined the neurogenic progenitors into intermediate transitional T1, T2 and T3 cell populations with the capacity to give rise to defined types of retinal neurons at specific developmental windows. Our analysis of 10–14 PCW samples demonstrated the presence of some of these neurogenic transitional progenitors; however, a better definition was obtained from the analysis of retinal samples (>10 PCW) due to a higher number of analysed cells within the retina per se (Fig. S1B, Supplementary Data 1). From 14 PCW, we were able to detect all types of retinal neurons apart from bipolar and Muller glia cells, which were detected from 14 and 16 PCW respectively (Fig. S1B,C, Supplementary Data 1). Published topographical studies report the highest density of microglia in the retinal periphery and retinal margin at ~8 PCW. Their presence in the central retina is not observed until 12 PCW (Diaz-Araya et al.[21]). In accordance with these studies, presence of cell clusters with high expression of microglia markers were observed from 14 PCW retinal samples (Fig. S1B, C, Supplementary Data 1).

### Transition of RPCs to T1-T3 neurogenic cell clusters

To identify gene regulatory networks (GRNs) that control RPCs specification and differentiation we integrated the transcriptomes of all retinal cells from 7.5–16 PCW human embryonic/foetal eyes with those of 10–21 PCW retinas. 48,856 cells were embedded using UMAP and clustered with Seurat (Fig. 1A). Forty-three clusters were identified: of these, thirty-eight were composed of RPCs and retinal neurons, one of microglia, one of fibroblasts and two expressed markers of more than one cell type (Supplementary Data 2). These last four clusters and an amacrine cell cluster with high expression of mitochondrial markers (cluster 41) were removed prior to further analysis. For illustrative purposes, all cell clusters defined as the same cell type, are annotated with the same colour (Fig. 1A).

Six RPC clusters were identified with enriched expression of typical markers including SFRP2, DAPL1, HES1, HMGA1, PAX6, RAX etc. Adjacent to the RPCs, a Muller glia cell cluster was found with high expression of RLBP1, SLC1A3, APOE, VIM, CLU, SOX9 (Supplementary Data 2). Eight clusters named proliferating RPCs were also found next to RPCs and were characterized by high expression of proliferation markers (e.g., MKi67, TOP2A) and RPCs markers (RAX, VSX2, SFRPP2, SOX2, HES1, ID3, HES6 etc.) (Fig. 1A, B). All differentiated retinal cell types were easily identified based on the expression of characteristic cell markers (Fig. 1B, Supplementary Data 2). Between the RPCs and retinal neurons, we identified a progenitor cell cluster with high expression of ATOH7, HES6 and DLL3, which was defined as the transitional T1 cluster. Two other cell clusters emanating from T1 were identified and defined as T2 and T3 based on the high expression of PTF1A and PRDM13 and FABP7 and OTX2 respectively (Supplementary Data 2).

To fully define the transitions from RPCs to T1, T2 and T3 neurogenic progenitors, we performed pseudo-temporal analysis of gene expression changes following cell cycle regression (Fig. 1C). This

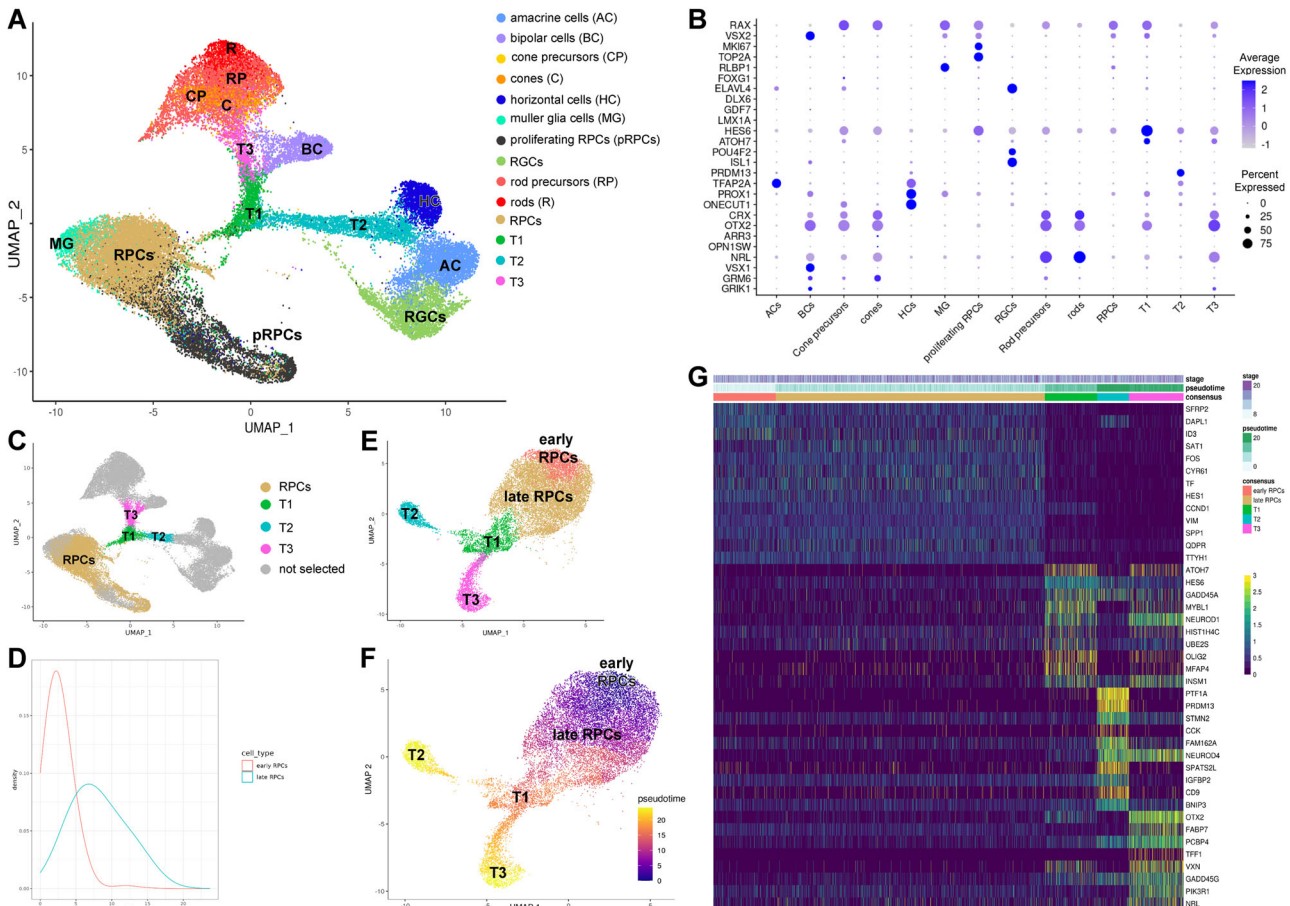

**Fig. 1 | RPCs identification and their developmental trajectories during human retinal development. A** UMAP plot of integrated scRNA-Seq foetal retina cells. Each cluster was identified based on expression of retinal specific cell markers. Highly expressed markers for each clusters are shown in Supplementary Data 2. **B** Dotplot showing highly expressed marker genes for each retinal cell type identified in the integrated scRNA-Seq data. **C** RPCs and transient neurogenic progenitors named T1, T2 and T3 were identified. Highly expressed markers for each cluster along the pseudotime trajectory are shown in Supplementary Data 2. **D** Density plot of RPC pseudotime scores showing a bimodal distribution corresponding to the early and late RPCs. **E, F** Pseudotime analysis demonstrating transition from early to late RPCs, and to T1 progenitors, which further commit to either T2 or T3 transient neurogenic progenitors. **G** Gene expression heatmap showing similarities in gene expression patterns between early and late RPCs, but distinct gene expression signatures in T1, T2 and T3 neurogenic progenitors.

analysis revealed bimodal densities of RPCs (Fig. 1D), mirroring the transition between early and late RPCs described by Lu and colleagues[5], followed by the transitional T1 and then T2 and T3 neurogenic progenitors (Fig. 1E, F). Early and late RPCs showed overall similar expression patterns of markers, but also some differences in the expression level. For example, the inhibitor of Wnt pathway (*SFRP2*) was more highly expressed in the early RPCs, while Muller glia cell markers (*VIM, FOS*) were more prominent in the late RPCs (Fig. 1G, Supplementary Data 2). High expression of well described neurogenic markers such as *ATOH7* and *HES6* was characteristic of the T1 and to a lesser extent of the T3 progenitor cluster (Fig. 1B). In accordance with the position of T2 between horizontal and amacrine cells, high expression of key transcription factors (*PRDM13, ONECUT1, TFAP2A*) required for differentiation to these two lineages was observed (Supplementary Data 2). The transitional cluster T3 shared the expression of highly expressed T1 markers, and additionally displayed the expression of typical photoreceptor precursors (*OTX2, CRX*) and bipolar cell markers (*VSX1*), consistent with its positioning between the T1, and photoreceptor and bipolar cell clusters (Fig. 1A–C, Supplementary Data 2). To tease out the transcriptional machinery that controls the specification of each differentiated retinal cell type and their transition, separate pseudo-temporal analyses of horizontal and amacrine, and photoreceptor and bipolar cells were conducted (Fig. S2, Supplementary Data 2). These analyses show that horizontal

and amacrine cells transit through a T1-T2 (Fig. S2A) and photoreceptor and bipolar cells through a T1-T3 transition state (Fig. S2B), corroborating recently published scRNA-Seq data on few stages of human foetal retinal development[6]. Although these pseudo-temporal analyses were performed on data subsets, the same results were obtained when the trajectory analysis was performed on the integrated scRNA-Seq UMAP (Fig. S2C).

During the course of these analyses many transcription factors highly expressed on defined progenitor subtypes or retinal neurons were identified. Human horizontal cells express the well-known markers such as *ONECUT1, ONECUT3, LHX1, PROX1* and show considerable overlap in gene expression with amacrine cells which are characterised by high expression of *TFAP2A, LHX9* and *MEIS2* (Fig. S2A, Supplementary Data 2). The cone and rod precursors were characterised by high expression of *THRB*, and *NRL* and *NR2E3* respectively, but interestingly also shared high expression of RGC and horizontal and amacrine cell markers (Fig. S2B, Supplementary Data 2). For example, high expression of RGC marker *SNCG* was found in the cone precursor cluster, while rod precursors displayed high expression of *PROX1*, a marker of retinal progenitors, horizontal and AII amacrine cells[22,23]. These findings were further corroborated by overlay expression plots (Fig. S3A), and immunofluorescence analysis showing co-expression of cone photoreceptor marker RXRG with RGC marker SNCG in the 8- 11 PCW retinal samples, which tailed off at 12 PCW (Fig. S3B). Notably, some of

these co-expressing cells were in a proliferative state at 8 PCW, but not in the later stages of development. Notably similar results were obtained in early stage (day 45) retinal organoids generated from human pluripotent stem cells[24], suggesting some "plasticity" with regard to TF expression in early stages of retinogenesis.

## ST analyses reveal the location of early RPCs in the CMZ

Our pseudo-temporal analysis above and a recent scRNA-seq analysis of human retinal development identified clear transcriptional signatures of early and late RPCs[5], however their spatial location in the developing retina has not been defined to date. To investigate this in detail, we performed ST analysis on an 8 PCW eye sample (Fig. 2A–D) revealing the presence of 12 clusters relating to cell-types with distinct spatial locations (Supplementary Data 3) including POM, lens, vitreous, corneal stroma, extraocular muscle and RPE (Fig. 2C, Fig. S3C). Notably, the optic stalk (cluster 9, Fig. 2C) was clearly defined by the ST analysis and characterised by high expression of *PAX2, ZIC1, HES1, SOX2, LHX2, THY1, PAX5* and *ID3* (Supplementary Data 3, Fig. S3C). Histologically, a single cell layer was present at the peripheral retina, but outer and inner neuroblastic layers were present in the central retina (Fig. 2A, C). The cells residing in the inner neuroblastic layer of central retina (cluster 5) were characterised by high expression of RGC markers such as *GAP43, PRPH, SNCG, INA, NEFL* and *NEFM* (Fig. S3C), corroborating our previous immunofluorescence findings[2].

High expression of genes typically marking the ciliary margin (e.g., *WNT2B*[25]), eye field (*RAX* and *PAX6*), early RPCs (*ID3, HMGA1, MDK, SFRP2*) and pigmented cell (*PMEL, TYR* and *TYRP1*) markers (Supplementary Data 3, Fig. S3C) were characteristic of the cluster 4, which was defined as CMZ. Subclustering of cluster 4 resulted in two further cell subclusters (Fig. 2E, F) with subcluster 0 expressing at high level marker genes associated with early RPCs (*FGF19, SFRP2, DAPL1, ZIC1, ID1, ID3, HMGA1, EEF1A1, TPT1, TMSB4X*), and subcluster 1 expressing at high level ciliary body (*KCNJ8, TPM2, ADGRA2A*) and iris pigment epithelium signature markers (*TYR, TRYP1, DCT, SILV, MLANA, PMEL*). In contrast, the progenitor cells residing in the outer neuroblastic layer of the central retina (cluster 10, Fig. 2C) were characterised by high expression of late RPC markers such as *RORB, CKB, HES1, HES5, ASCL1, HES6* and *NEUROD1* (Supplementary Data 3). This led us to hypothesise that the early and late RPCs could be spatially located in different regions of the developing human retina. To investigate this further, we generated gene expression signatures for early and late RPCs, T1, T2 and T3 neurogenic progenitors based on the differentially expressed gene markers between the different cell types defined in our subsets used for the pseudo-temporal analysis (Fig. 1F, G). These gene expression signatures were further supplemented with marker genes from recent published studies[2,5,20] (Supplementary Data 4). Violin plots (Fig. 2G, I) indicated the highest aggregate expression score of early RPCs markers in the subcluster 0 of CMZ (cluster 4), while the highest

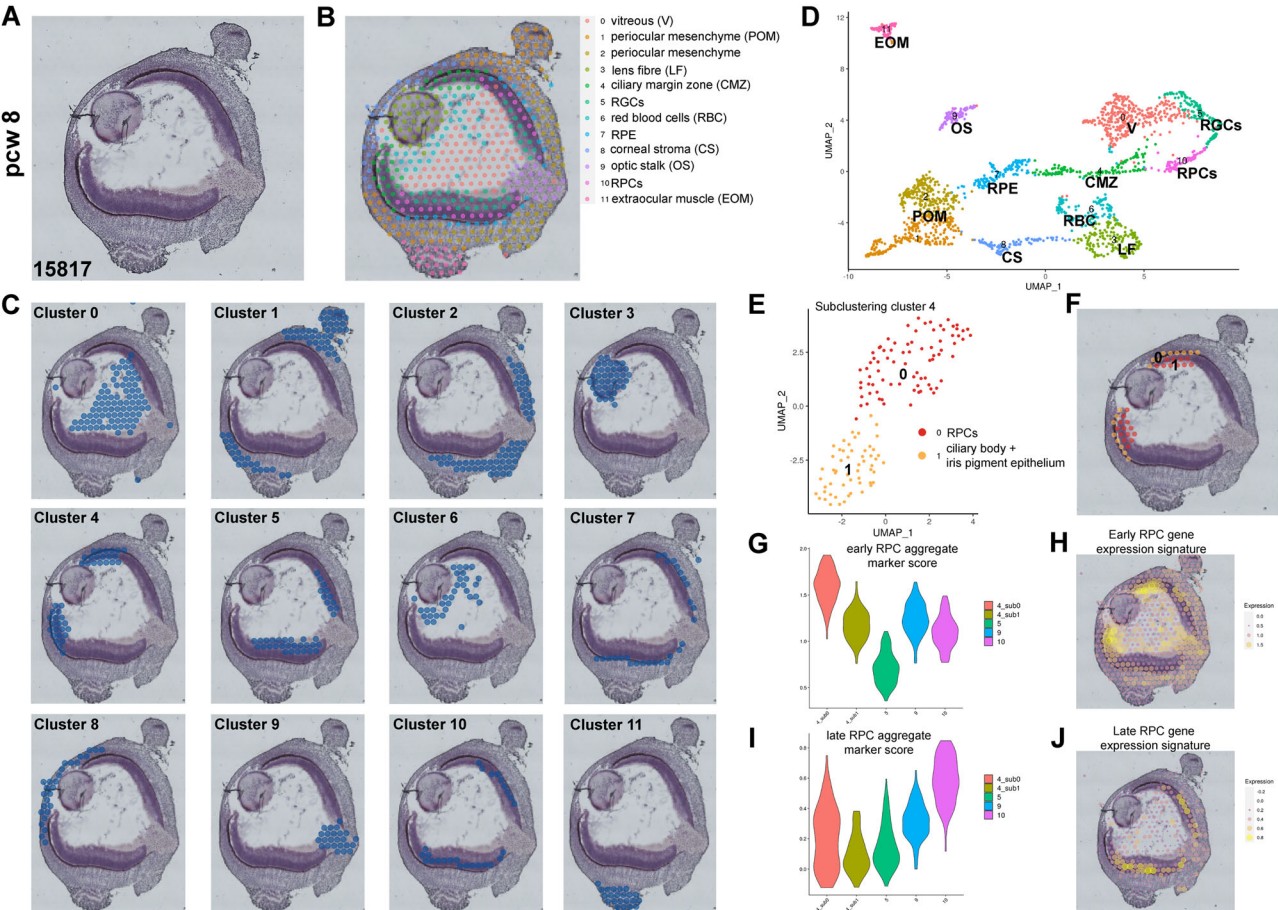

**Fig. 2 | ST analysis of 8 PCW human eye sections reveals the location of early RPCs in the CMZ. A** Representative histological staining of the 8 PCW fresh frozen human eye section. Four sections from the same eye sample were processed for ST analyses. **B, C** Spatial localisation of the 12 clusters identified from the ST analysis. Highly expressed markers for each cluster are shown in Supplementary Data 3. **D** UMAP of spatial transcriptomics scRNA-Seq data. **E** Subclustering of ciliary margin zone (cluster 4) reveals the presence of two subclusters namely RPCs and

ciliary body, and iris pigmented epithelial cells: their spatial localisation is shown in panel (**F**). **G** and (**I**) Expression violin plots showing the highest aggregate expression scores for early RPCs in the peripheral retina (CMZ) and late RPCs in the central retina respectively, compared to all other retinal clusters identified in the ST analysis. **H** and (**J**) Early and late RPC gene expression signatures superimposed on the spatial image of 8 PCW human eye.

expression score of the late RPCs markers was observed in the outer neuroblastic layer of central retina (cluster 10). These findings were further corroborated by the spatial mapping of progenitor cells with gene signatures defined in Supplementary Data 4, showing the predominant localisation of early RPCs in the CMZ, the late RPCs in the central retina (Fig. 2H, J), and the transitional neurogenic clusters T1, T2 and T3 predominantly in the outer neuroblastic layer of the central retina (cluster 10, Fig. S3D). Interestingly, an early RPCs expression signature was also observed in and around the optic stalk (Fig. 2H), albeit less intense than in the CMZ.

Similar cell clusters with enriched expression of RPCs and pigmented epithelial cell markers were consistently found in the ciliary margin zone of 10, 11 and 13 PCW eye specimens analysed with the ST approach (Fig. 3A–E, Supplementary Data 3). To perform a comparative assessment of RPCs between these early developmental stages, spots within the clusters annotated as CMZ in ST were selected. The Seurat function "AddModuleScore" was used to the calculate an aggregate expression score of the early RPC marker genes ("*FGF19*", "*SFRP2*", "*DAPL1*", "*ZIC1*", "*ID3*", "*HMGA1*", "*EEF1A1*", "*TPT1*", "*TMSB4X*", "*MDK*", "*FOXP1*", "*GN2BL1*", "*HNRNPA1*", Supplementary Data 4) for each spot within the CMZ region. The aggregate scores were then plotted as

a violin plot. There was higher aggregate expression of early RPC marker genes in the CMZ in 8 PCW compared to 10, 11 or 13 PCW (Fig. 3F). A similar analysis for the late RPC markers, showed no changes from 8 – 13 PCW specimens (Fig. 3G). We were unable to fit larger size eyes to the current Visium ST slides, thus it was not possible to extend the ST analyses to foetal eyes older than 13 PCW. However, we utilised the scRNA-Seq data and plotted the ratio of early to late RPCs, showing a significant reduction in fraction of early RPCs from 8 PCW onwards (Fig. 3H).

To complement the ST analyses and obtain more insights into early and late RPC localisation into the CMZ during retinal development, we performed RNA-Scope investigations using a marker of early RPCs (*ZIC1*), iris pigmented epithelium (*TFPI2*), late RPCs/neurogenic progenitors marker (*HES6*) and ciliary body (*OPTC*) in eye samples from 6.3-16 PCW. Consistently with data obtained above, we observed *ZIC1* expression in the CMZ as well as rest of retinal neuroepithelium of 6.6 PCW retinas (Fig. S4A, B). In contrast, the expression of *HES6* was first seen in the central retina (Fig. S4B) spreading to the periphery, up to CMZ, from 8 PCW until the last 16 PCW specimens examined (Fig. S4D, G, Fig. S5C, D, G). Although *ZIC1* expression was still present in the CMZ of 8 PCW retinas (Fig. S4C, D) it was reduced and by 10

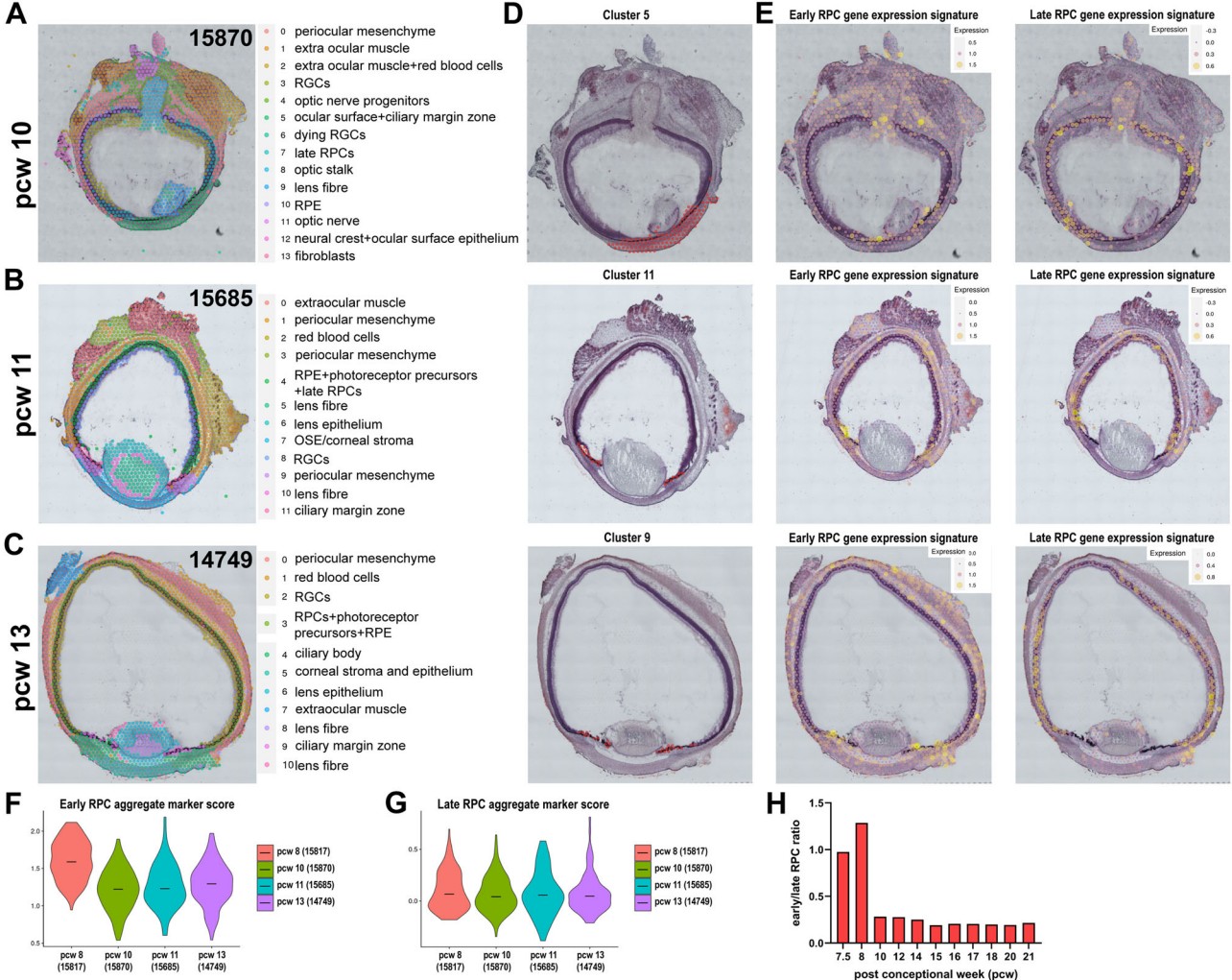

**Fig. 3 | ST analyses demonstrate decreased presence of early RPCs in the ciliary margin zone as development proceeds from 10 to 13 PCW. A–C** Spatial localisation of all cell clusters identified from the ST analyses. Four sections from each eye sample were processed for ST analyses. **D** The CMZ clusters are shown for each developmental stage. **E** Early and late RPC gene expression signatures superimposed to the spatial images of each eye specimen. **F**, **G** Violin plots showing early (**F**) and late (**G**) RPCs aggregate expression scores across four developmental stages, demonstrating the reduction of early RPCs during development. **H** Early RPCs reach a peak at 7.5-8 PCW and then decline from 10 PCW onwards in the human developing retina. The ratio of early to late RPCs is inferred from the scRNA-Seq data.

PCW, it was localised in small patches at the posterior end of CMZ of <50% of eye sections assessed (Fig. S4E, F, G). In 13 and 16 PCW, *ZIC1* expression was completely absent from the CMZ (Fig. S5A–F), however its expression in the rest of retinal neuroepithelium persisted in a co-expressing pattern with *HES6* (Fig. S5C, D, G). Together these data corroborate localisation of early RPCs in the CMZ of developing human retinas with decreasing frequency from 8 PCW of development and reveal the propagation of neural retina differentiation from the centre to the periphery.

### Characterisation of chromatin accessibility

To investigate chromatin accessibility during human retinal development, we performed scATAC-Seq analyses of two developing human eyes (8.5 PCW), and ten retinal samples dissected from 10–21 PCW (Supplementary Data 5). 118,883 cells were captured using the 10XGenomics Chromium Single Cell ATAC Library and Gel Bead Kit. The corresponding scRNA-Seq datasets were used as reference maps to identify ATAC-Seq clusters for each sample (Fig. S6). Similarly, to scRNA-Seq analysis, defined clusters of corneal stroma, epithelium, endothelium, and periocular mesenchyme, CMZ, extraocular muscle, red blood cells, microglia and optic nerve were found in the 8 PCW human eye samples alongside retinal clusters comprised of RPCs, RGCs, horizontal and amacrine cells and transient neurogenic cluster T1 (Fig. S6A). All the retinal cell types were present between 10–14 PCW, albeit some at the precursor stage (for example rod precursors at 10 PCW), in addition to RPCs and the three types of transient neurogenic clusters T1, T2, T3 (Fig. S6A). From 16 PCW, small clusters of microglia and the last-born cell type, Muller glia cells were evident

(Fig. S6B), corroborating the scRNA-Seq data and previously published sequential order of retinal cell emergence[2,5,6].

54,045 retinal cells from 12 samples and 9 developmental time-points encompassing 8–21 PCW with a 3694,28 average median fragments per cell were integrated. The cells were clustered using the chromatin accessibility peaks near previously known marker genes resulting in 22 clusters (Fig. S6C, Supplementary Data 5). These included the abundant cell types (for example rods, cones) as well as the rarer cell types (e.g., microglia) and several types of amacrine cells (gabaergic, glycinergic and starburst). The DNA accessibility peaks were classified using annotation from cellranger and associated with promoters (if found within -1000 – +100 bp of the transcription start sites), exons, introns, distal (if found within 200 kb of the closest transcription start site), or intergenic regions (if not mapped to any genes) (Fig. 4A, Supplementary Data 6). This analysis enabled us to identify cell type specific regions of accessibility in RPCs, transient neurogenic progenitors T1, T2, T3 and retinal neurons (Fig. 4B). All clusters displayed scATAC-Seq marker peak enrichment of cell type specific marker genes (Fig. 4C).

We then went on to predict transcription factor (TF) binding motifs within the scATAC peaks using Signac (Fig. 5A, Supplementary Data 7), followed by foot printing validation analysis (Fig. 5B, C). In RPCs, we identified binding motifs for TFs expressed in the optic stalk (VAX1, VAX2[26]), eye field (RAX[27]) and RPCs (LHX6, VSX2, SOX2[28–30]) as well as TF binding motifs that were shared with other cell types (for example FOS, NFI family members and SOX6 binding motifs were shared with Muller glia cells, Fig. 5A). Notably we identified binding motifs for TFs not previously associated with RPCs, for

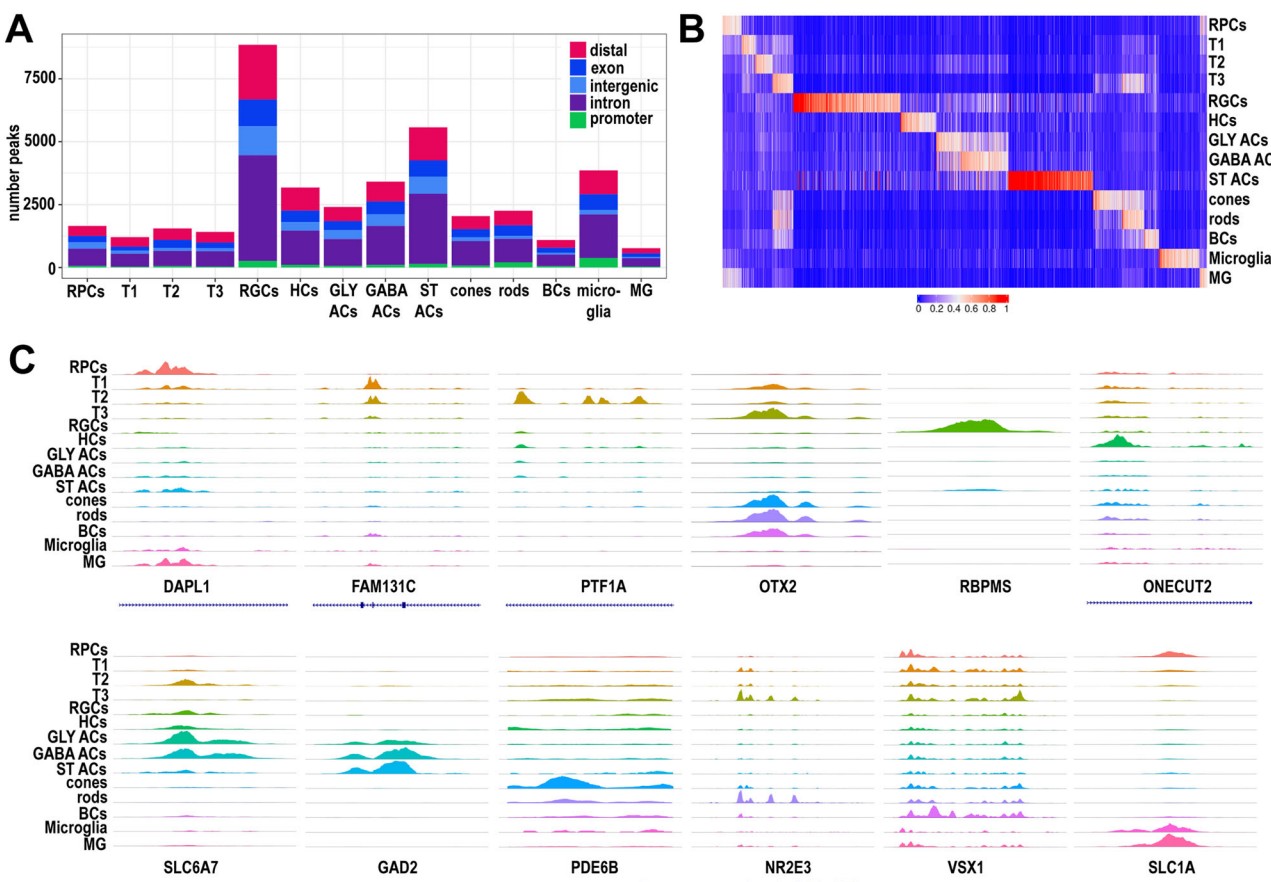

**Fig. 4 | Single cell ATAC-Seq analysis of developing retina samples reveals cell type specific chromatin accessibility profiles. A** The number and type of chromatin accessibility profiles for each cell type. **B** Heatmap showing differentially accessible of chromatin accessibility peaks (columns) for each cell type (rows). **C** Representative examples of chromatin accessibility peaks for retinal cell specific marker genes. Each track represents the aggregate scATAC signal of all cells from the given cell type normalized by the total number of reads in TSS regions.

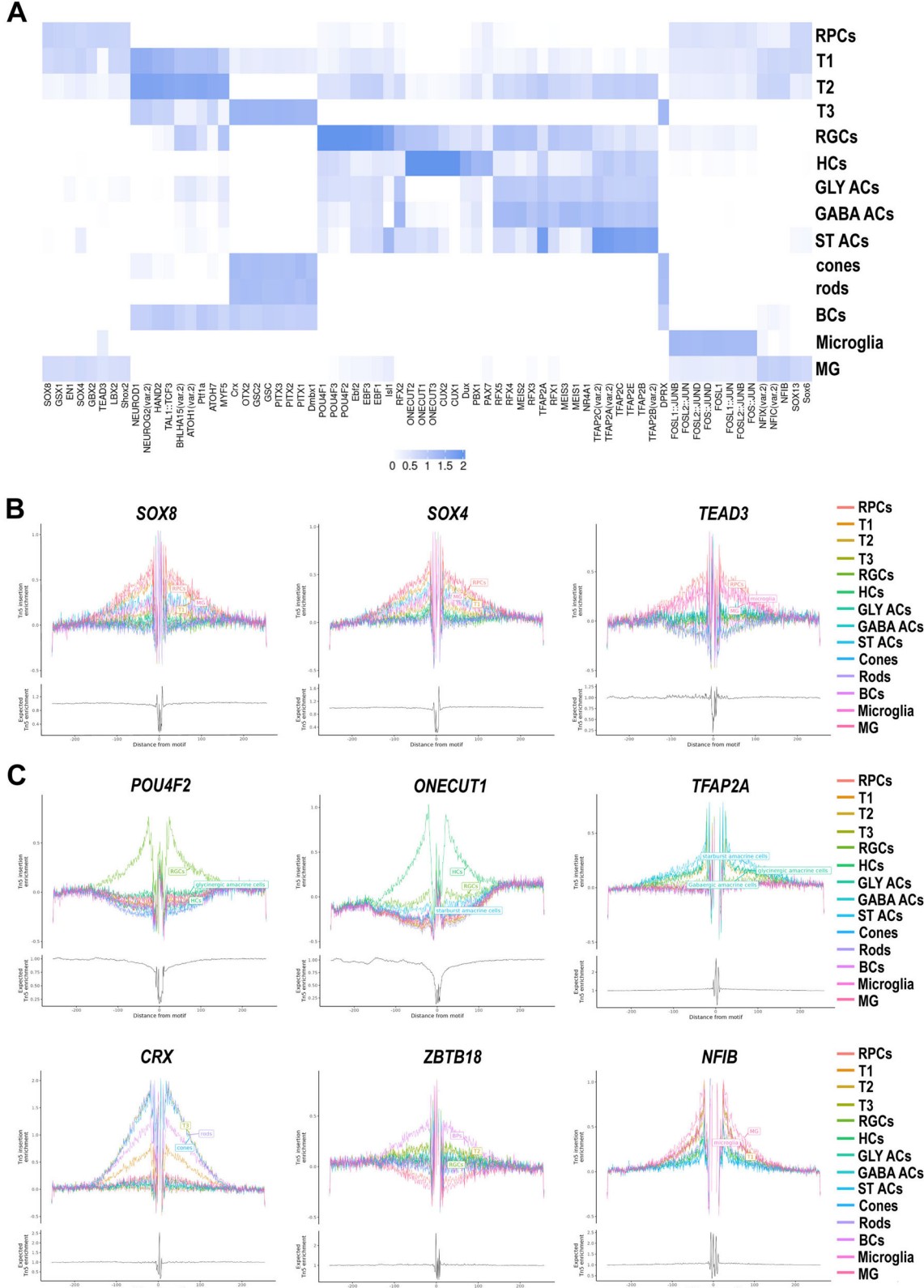

**Fig. 5 | Motif analysis of accessible DNA peaks predicts cell type specific TFs in the developing human retina. A** Heatmap of transcription factor binding motifs enriched in each cell type. More significant enrichment is indicated by the darker colours. **B** Footprinting analysis of selected TFs predicted to show a significant enrichment in RPCs. **C** Footprinting analysis of selected TFs predicted to show a significant enrichment in transient neurogenic progenitors and retinal neurons. Additional abbreviations to those mentioned in the main text: Gly ACs glycinergic amacrine cells, GABA ACs gabaergic amacrine cells, ST ACs starburst amacrine cells, HCs horizontal cells, MG Muller glia cells.

example EN1, GBX2, LBX2, SHOX2 and GSX1 (Fig. 5A, B, Supplementary Data 7).

In accordance with sequential emergence of T2 and T3 neurogenic clusters from T1, we observed a set of shared TF binding motifs for NEUROG2, NEUROD1, HAND2, TAL1:TCF3, PITF1A, ATOH7, BHLHA15, MYF5 and ATOH1 (Fig. 5A). The earliest born retinal cell types, RGCs displayed the POU4F and EBF family members binding motifs, whilst horizontal and amacrine cells were characterised by ONECUT and CUX, and TFAP2 and MEIS family members binding motifs respectively (Fig. 5A, C). These findings were largely corroborated with the SCENIC+ analysis (Fig. S7A). Rod and cone photoreceptors are derived from the T3 neurogenic progenitors; hence shared binding motifs were identified for TFs such as OTX2[31], CRX[32] and DMBX1[33] (Fig. 5A, C, Fig. S7A), which are well described in the literature for their role in photoreceptor specification. Notably, shared binding motifs in T3 and photoreceptors were also highlighted for PITX1 and GSC TFs not associated previously with a role in rods or cones (Fig. 5A), but those were not reproducible using SCENIC + . Since PITX1 has a binding motif that is extremely similar to OTX2[34], a TF that plays a key role in defining the T3 differentiation to photoreceptor and bipolar cells[31,35], it is likely that the results we obtained are due to the similarity in the binding motifs between these two TFs. Moreover *PITX1* was expressed in a minute fraction of cells identified along the T3-photoreceptors lineage (0% of cone precursors, 0.17% of cones, 0.3% of rod precursors, 0.82% of rods and 0.1% of T3 progenitors), hence its potential role as regulatory TF in photoreceptor specification is unlikely. Bipolar cells are also derived from the T3 progenitors; hence enrichment of T3 TF binding motifs (such as OTX2) was observed in addition to less well characterised TFs such ZBTB18 (Fig. 5C). Enrichment of NFI binding motif family members was observed for glia cells (Muller glia and microglia) in accordance with their role in regulating specification of the late-born cell types in the retina[36].

## scATAC-Seq enables prediction of gene regulatory networks
To identify GNRs governing the RPCs differentiation to transient neurogenic progenitors (T1, T2, T3) and the retinal neurons, the upstream regulator tool in IPA was used, combined with overlay analysis of DA peaks. A large number of upstream regulators including TGFβ, IGF-1, FGF-2, Sonic hedgehog were predicted to be activated in RPCs (Supplementary Data 8). Their main common characteristic was predicted activation of key genes encoding proteins important for RPCs proliferation (*HES1*[37], *CCND1*[38], *ID3*[39]) and inhibition of transcription factors (*ATOH7*[40], *NEUROD4*, *NEUROD1*[41], *PTF1A*[42], and *HES6*[43]) (Fig. 6A, B) that define the RPC competence to RGCs, horizontal and amacrine cell fates, and photoreceptor differentiation. Amongst the predicted inhibited upstream regulators, we found key transcription factors such as *PAX6*[44] and *ASCL1*[45], shown to control the timing and specificity of retinal neurogenesis (Fig. 6C, D). Together these findings suggest the presence of a finely tuned balance between activation of proliferation regulators and inhibition of neurogenic cues to maintain RPC self-renewal. Conversely in the transient neurogenic progenitors, we predicted the activation of upstream neurogenic regulators (e.g., ASCL1 in T1, FOXA2 in T2, GTF2IRD1 in T3), which control the expression of TF necessary for retinal cell type specific differentiation (Fig. S8A–C) and inhibition of regulators (e.g. BMP4, LIN28A, IGF-1) that govern RPCs fate (Fig. S8D–F).

Performing the same analysis on the differentiated retinal cells revealed some interesting insights and predicted upstream regulators (Supplementary Data 8). For example, Eomesodermin (EOMES), a target gene of Pou4f2, required for RGC and optic nerve development in mouse[46], was predicted to be activated in horizontal cells (Fig. S9B), resulting in activation of *LHX1*[28,47], a transcription factor that specifies horizontal cells, while suppressing *LHX9*, a transcription factor required for amacrine cell subtype specification[28]. We identified a putative upstream regulator in RGCs, namely KLF2 (Fig. S9A), which is

predicted to inactivate Notch signalling, an important event required for RGC differentiation[48]. Our scRNA-Seq data have shown that the basic-helix-loop-helix PTF1A is highly expressed in the T2 progenitors, which give rise to amacrine and horizontal cells. This central role was beautifully highlighted by the SCENIC+ analysis (Fig. S7B), which identified PTF1A as upstream regulator of two key TFs namely ONECUT2 and LHX9 governing horizontal and amacrine cell genesis respectively, with the latter regulation being mediated by CCND1. IPA analyses corroborated these findings, identifying PTF1A as key upstream regulator in amacrine cells, resulting in activation of TFs that promote amacrine cell specification and function (Fig. S9C). OTX2 and CRX transcription factors, were also identified as key upstream regulators of multiple genes involved in cone or rod photoreceptor differentiation (e.g., *RXRG*, *NR2E3*), outer segment formation (e.g., *PROM1*, *PRPH2*), connecting cilia (e.g., *RP1*) and phototransduction (e.g., *PDE6B*) (Fig. S7C).

Our analysis was also able to predict more complex upstream regulators, which are ligand-activated and able to play a role in the control of gene expression in a cell type specific manner. For example, the peroxisome proliferator-activated receptor γ (PPAR*G*), was identified as a putative upstream regulator in cone photoreceptors (Fig. S9D) and predicted to activate amongst others *PRDM1*, which has been demonstrated to stabilise photoreceptor cell fate in OTX2+ progenitors by preventing bipolar cell induction[49]. In contrast, the upstream regulator TCF7 in bipolar cells is predicted to suppress *PRDM1* and activate expression of key genes important for bipolar cell function such as *GNAO1* and *TRPM1*[50] (Fig. S9F). G-protein coupled receptors (GPCRs) play a significant role in many tissues by transducing complex signalling networks that coordinate gene expression. In accordance, one of the putative activated upstream regulators in rods, was the GPCR Rhodopsin (RHO), the most abundant protein in rods which functions as the primary photoreceptor molecule of vision (Fig. S9E). Our data suggests that RHO may regulate the transcription of rod specific phosphodiesterases (e.g., *PDE6G, PDE6B*) and transducins (*GNAT1, GNGT1*). This data driven hypothesis is corroborated by published evidence demonstrating that human retinal organoids and mice with altered expression of *Rho* display changes in expression of phototransduction genes[51–53].

## TEAD TFs play an important role in RPC proliferation
The TF binding motif analysis in RPCs (Fig. 5A) revealed an enrichment for TEA domain (TEAD) TFs in RPCs (Fig. 7A, Supplementary Data 7). TEADs play an essential role in mediating YAP-dependent gene expression resulting in transcription of target genes responsible for cell proliferation, inhibition of apoptosis or retinal neurogenesis[54]. During retinal development, YAP's expression is restricted to the outer neuroblastic layer where RPCs reside[55]. The TEAD TFs display different expression patterns in the retina, with Tead2 being highly expressed in the proliferating cells located at the basal side of the outer neuroblastic layer of mouse retina and Tead3 being highly enriched in the inner neuroblastic layer (genepaint.org).

To assess if TEADs binding is essential for RPC development and differentiation, we treated PSCs-derived retinal organoids with TEAD palmitoylation inhibitor MGH-CP1, that blocks TEAD2/4-YAP interaction and suppresses the expression of their target genes in cancer cell lines[56]. Treatment was started at day 25, coinciding with immunofluorescence detection of RPCs in the retinal organoids. Three different doses (2.5, 5 and 10 μM) of MGH-CP1 were added every 2 days to the culture media for 14 days. Under control conditions (treated with vehicle) retinal organoids develop a bright phase retinal neuroepithelium by day 28 (day 3 of treatment) which expanded over time. By the end of treatment the majority of control retinal organoids (>75%) had full coverage with bright phase retinal neuroepithelium, with a minority of organoids displayed partial coverage (15%) or no coverage (7%) at all (Fig. S10A, B). In contrast, the great majority of retinal

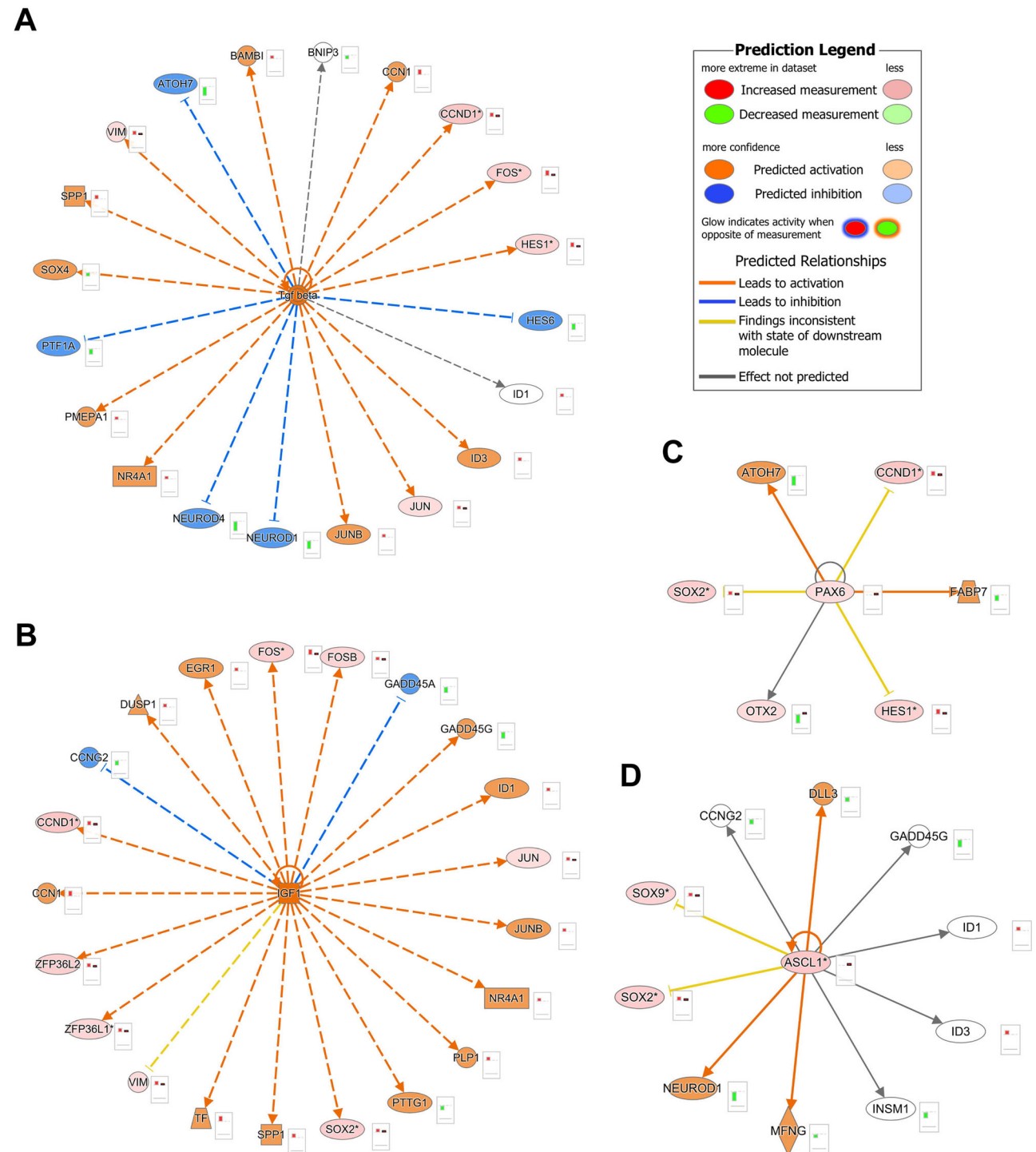

**Fig. 6 | Gene regulatory networks in RPCs.** Representative gene regulatory networks in RPCs depicting activated upstream regulators (**A**, **B**) and inhibited upstream regulators (**C**, **D**) and their target genes. Upstream regulatory networks were generated with IPA using differentially expressed genes from the scRNA-Seq data and differential accessibility analysis in the scATAC-Seq data. The networks show predictions of upstream regulators which might be activated or inhibited to explain observed upregulation/downregulations in the data. The barplots next to each molecule represent the relative expression in the sRNA-Seq (column 1) and scATAC-Seq datasets (column 2). The colours for the network nodes/barplots indicate observed upregulation/ increased chromatin accessibility (red), predicted upregulation/increased chromatin accessibility (orange), observed downregulation (green) and predicted downregulation/ decreased chromatin accessibility (blue). The colour of the edges represents the relationships between the molecules; orange = prediction and observation are consistent with activation; blue = prediction and observation are consistent with downregulation; yellow = prediction and observation are inconsistent; and grey relationship between the molecules is available in the IPA knowledge database. *- indicates duplicates in scATAC-Seq dataset.

organoids treated with MGH-CP1, displayed either partial or full loss of bright phase neuroepithelium in a dose dependent manner. These findings were fully corroborated by immunofluorescence analyses, which showed the presence of partial retinal neuroepithelium harbouring VSX2⁺ RPCs in organoids treated with 5 μM MGH-CP1 and much reduced and mislocalised RPCs in the organoids treated with 10 μM MGH-CP1 (Fig. 7B, F). The latter were also often characterised by the presence of a few mislocalised SNCG+ RGCs in the apical layer of

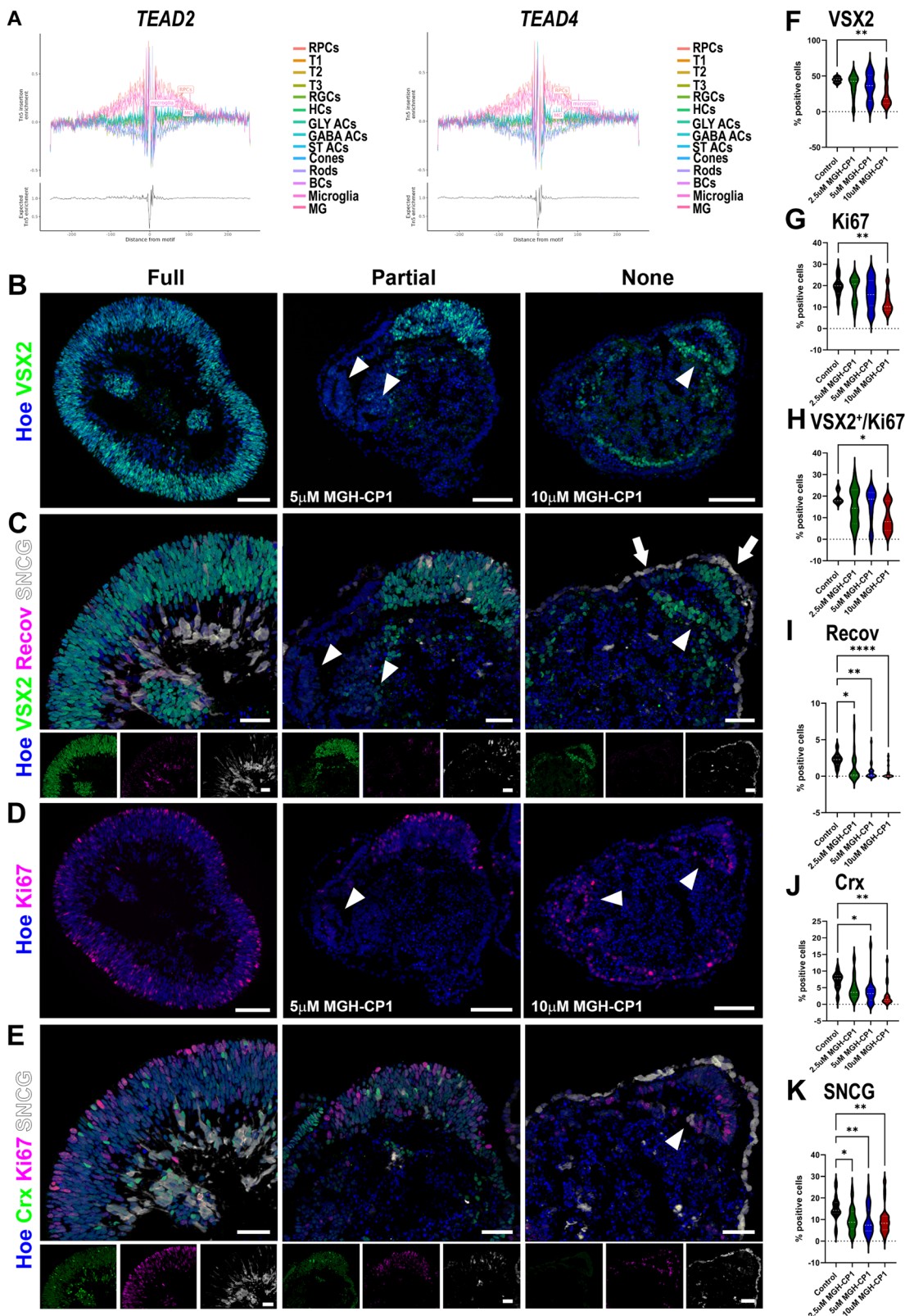

the organoids and internal rosettes with VSX2⁺ RPCs or Ki67 proliferating cells (Fig. 7C–E). Loss of retinal identity in MGH-CP1 treated organoids was corroborated by bulk RNA-Seq differential gene expression analysis revealing downregulation of optic vesicle (*TWIST1*)[57], RPC (*NDUFA4L2*)[58], amacrine (*TFAP2B*)[59], bipolar (*IRX5*)[60] and Muller Glia (*CRYGC*) cell markers (Supplementary Data 9), and quantitative immunofluorescence showing attenuated RPCs,

photoreceptor, and RGCs specification (Fig. 7F, I–K). In accordance with TEADs role in activating transcription of genes important for cell proliferation, we noted a significant reduction in the fraction of Ki67⁺ and cycling RPCs (Ki67⁺VSX2⁺) in the retinal organoids treated with the highest dose of MGH-CP1 (Fig. 7G, H), and a small increase in the percentage of Caspase 3+ apoptotic cells (Fig. S10C). No significant difference across conditions was observed in the percentage of

**Fig. 7 | TEAD binding plays a significant role in RPC proliferation. A** Footprinting analysis of TEAD2 and TEAD4 showing a significant enrichment in RPCs. Additional abbreviations to those mentioned in the main text: Gly ACs- glycinergic amacrine cells, GABA ACs – gabaergic amacrine cells, ST ACs – starburst amacrine cells, HCs – horizontal cells, MG- Muller glia cells. **B–K** Quantitative immunofluorescence analyses for the presence of VSX2$^+$ RPCs (**B, F**), Ki67$^+$ proliferating cells (**D, E, G**), VSX2$^+$Ki67$^+$ (**H**), SCNG$^+$ RGCs (**C, E, K**), and Recoverin$^+$ (**C, I**) and CRX$^+$ photoreceptor precursors (**E, J**) reveal loss of RPCs, disturbed retinal lamination, and attenuation of photoreceptor and RGCs specification. Bottom panel insets at (**C**) and (**E**) panels show individual antibody and nuclear staining. White arrowheads show the presence of rosettes comprised of RPCs or Ki67$^+$ proliferating cells. Scale bars 100 μM for (**B–D**) and 50 μM for (**C–E**) and bottom inset panels. **F–K** Data presented as median and quartiles. 9 – 39 retinal organoids per condition were used as documented in the Source data file. One-way ANOVA (**F–H**) or Kruskal-Wallis (**I–K**) with Dunnett's multiple comparisons test (*$p < 0.05$; **$p < 0.01$; ***$p < 0.001$). Source data are provided as a Source Data file.

apoptotic RPCs (Fig. S10C). Together these data suggest that loss of retinal identity in MGH-CP1 treated organoids is most likely due to attenuated RPC proliferation.

## Discussion

In this study we analyzed the transcriptional and DNA accessibility profiles of over 277,000 cells from 35 human eyes and retinal samples collected from 7.5-21 PCW of human development to delineate the diversification of RPCs and the GRNs that govern their differentiation. Using spatiotemporal single cell RNA-Seq analyses we demonstrate the transient localisation of early RPCs in the CMZ of developing human retinas and propagation of neural retina differentiation from the centre to the periphery. Single cell ATAC-Seq analysis revealed a significant enrichment of TEAD transcription factor binding motifs in RPCs, which when inhibited led to loss of retinal lamination, and retinal identity due to reduced RPC proliferation.

The human eye is a heterogenous entity, comprised of diverse tissues derived from neuroectoderm, neural crest and mesenchymal cells. Single cell RNA-Seq studies have focused on the transcriptome of developing and/or adult cornea[61], iris, ciliary body[62], neural retina[5,6,63], RPE and choroid[64,65]. However, a comprehensive single cell spatiotemporal profiling of human developing eyes has not been attempted before. Herein, we undertook scRNA-Seq analysis of 10 developing human eyes encompassing 7.5-10 PCW of development, revealing the transcriptional signature of neural crest, extraocular muscle and POM cells in addition to ciliary body, iris pigmented epithelium, ocular surface epithelium, stroma and endothelium, neural retina and RPE cells. In all cases, the neural crest cell clusters were closely associated with cell types derived therefrom including corneal stroma and endothelium, POM, periocular connective tissue and melanocytes, or those that require early signals from neural crest for their development such as extraocular muscle[66] and lens fibres[67]. Defects in neural crest formation are at the heart of several severe craniofacial and ocular anomalies including ocular coloboma, glaucoma etc[17]., hence a comprehensive understanding of transcriptome of neural crest cells and its derivatives as described herein, is of great importance for understanding the complexities underlying congenital eye diseases.

Taking advantage of ST, we were able to spatially locate various structures in the developing human eyes of 8–13 PCW, revealing the spatial and single cell transcriptome of the optic stalk, which forms the primitive connection between the retina and the brain. The region around the optic stalk is initially continuous with the optic fissure margins. In the human eye fissure closure begins in the middle and most of the posterior part eventually forms part of the optic nerve head. Like the CMZ, the fissure margins are a zone of transition between the neural retina and RPE, thus it is likely that the optic stalk region may retain characteristics in common with the CMZ. In accordance, high expression of early RPC markers (*SOX2, HES1, ZIC1, NR2F1, LHX2*) and typical optic stalk marker PAX2, was observed in the optic stalk cluster (Supplementary Data 3). Importantly, the ST approach enabled us to analyse at single cell resolution the CMZ in developing human eyes of 8, 10, 11 and 13 PCW. Using human PSCs-derived retinal organoids, Kuwahara and colleagues provided evidence of a putative CMZ, containing sphere-forming cells, able to generate de novo retinal cells[68]. However, whether the human CMZ is a source of RPCs during embryogenesis has been a long-standing question in the field. Using a combination of single cell RNA-Seq and ST, we show the presence and localisation of early RPCs in the CMZ of 8 PCW human eyes with decreasing frequency as development proceeds from 8 PCW onwards. Moreover, the CMZ located early RPCs were characterised by distinct expression signature compared to the ciliary body and iris pigmented epithelium, suggesting that they are a distinct entity to the putative stem cell ciliary body cells suggested by Gautam and colleagues[62]. Together our data provide evidence on the transitional presence of early RPCs in the CMZ at a particular stage of human retinal development that needs to be functionally validated both in vivo using a larger number of early eye specimens and in vitro in retinal organoids by combining cell barcoding with single cell analyses.

In fish and amphibians, only the most central part of the retina is formed during embryogenesis with the rest of retina formed by addition of new neurons from the CMZ, a process which continues throughout life in fish. In contrast, avian retinogenesis occurs nearly exclusively during embryogenesis. The differentiation of retinal cells begins in central retina and proceeds towards the periphery, however retinal neurogenesis also occurs in the far peripheral retina (CMZ) of embryonic and post-hatch chick[69]. In mammals, the retina follows a pattern of central to peripheral growth and once the retinogenesis is complete there is no continued generation of retinal neurons. Studies performed in prenatal mice have however shown that Msx1+ expressing cells in the CMZ give rise to RPCs, which proliferate and generate all the retinal cell types, indicating two modes of neurogenesis: direct from neurogenic RPCs and indirect from CMZ-derived progenitors[16]. Our study provides molecular evidence for the transient presence of early RPCs in the ciliary margin of human developing retina as well as central to peripheral mode of neurogenesis. How could these findings be reconciled? Are the early RPCs in the CMZ the only RPCs present during embryogenesis? Our spatial transcriptomic analyses of early human embryos (8 and 10 PCW) shows an early RPC molecular signature also in the central retina, suggesting that two sources of early RPCs may be present transiently in human embryos similarly to prenatal mice. The relationship between CMZ and centrally located early RPCs cannot be answered from our current data, but given the reports of CMZ-like presence in human retinal organoids[68], cell lineage tracing can be performed in this system through cell barcoding and single cell analyses.

Through pseudo-temporal analyses we were able to show that early RPCs are able to differentiate into late RPCs, which are localised throughout the outer neuroblastic layer of developing retina, apart from the CMZ. Those give rise to transient neurogenic progenitors, named T1, T2, and T3, which were identified by both scRNA- and ATAC-seq, corroborating recently published data[6]. The pseudotime analyses demonstrated that horizontal and amacrine cells via T1, T2 progenitors and bipolar cells and photoreceptors via T1, T3 progenitors. We were also able to demonstrate that both cone and rod photoreceptors develop from a precursor stage which precedes their maturation. Notably, we also observed at the early stages of development some "plasticity" regarding lineage transcription factor expression such as *SNCG*, a marker of RGCs was expressed in cone precursors and *PROX1*, a marker of horizontal and amacrine cells expressed in rod photoreceptors. Strikingly, Prox1 expression has been noted in rod progenitors as well as the Müller glia cells in the fish[70], while high coexpression of RGC markers was reported recently in the proliferating

cones of retinoblastoma pluripotent stem cell derived organoids as well as patient retinoblastoma samples[71,72]. Together these data suggest that although the expression of certain TFs maps to individual retinal cell trajectories, some of these may be reused temporary during the development of other retinal cell types or redeployed upon re-entry into the cell cycle and conversion to malignancy.

We complemented the scRNA-Seq with ATAC-Seq analysis of similarly staged developing human eyes and retinas. We found the scATAC-Seq to be more informative when it came to subclustering of amacrine cells into the three main subtypes namely gabaergic, glycinergic and starburst amacrine cells. Notably we also found scATAC-Seq to predict cell type clusters at earlier stages than scRNA-Seq analysis. For example, horizontal and amacrine cell precursors were identified at 8 PCW in the scATAC-Seq but 10 PCW in the scRNA-Seq. This could be due to DNA chromatin accessibility preceding key TF expression and cell fate determination. Similarly to recently published studies of scATAC-Seq in the developing and adult retina[7,10–12], we were able to reveal the accessible chromatin regions and putative TF binding motifs for the RPCs, transient neurogenic progenitors and all the retinal cells as they emerged during the course of development. This identified both well characterised TFs (e.g., POU4F in RGCs, ONECUT2 in horizontal cells, RAX in RPCs) as well as TFs (e.g. EN1, GBX2, LBX2, SHOX2, TEAD1-3 and GSX1 in RPCs), which deserve further functional validation in animal models and retinal organoids.

The scATAC-seq analyses enabled us to predict GRNs and TFs that govern retinal neurogenesis. In particular, enrichment of TEAD binding in RPCs was highlighted, suggesting a role for the Yap-Hippo signalling pathway during human retinal development. Evidence demonstrating the role of Hippo-Yap signalling during retinogenesis in mice has shown that developing retinas devoid of Yap display disrupted apical junctions, rosette formation and loss of laminar arrangements[53]. Our data fully corroborate these findings, showing the presence of rosettes within retinal organoids and disrupted laminar organisation when TEAD binding is disrupted. In addition to playing a role in retinal lamination, activated Yap has been shown to simultaneously interact with TEADs and Rx1 during zebrafish embryogenesis to drive the expression of proliferation-related genes and attenuate the trans-activation of photoreceptor genes respectively[73]. In accordance we observed a significantly reduced fraction of cycling RPCs, and decreased gene or protein expression for all retinal cell types, indicating loss of retinal identity due to a stall in RPC proliferation. Furthermore, we observed attenuated photoreceptor differentiation, which suggests that upon inhibition of TEAD binding, Yap may interact with the human ortholog of Rx1 (RAX) to suppress photoreceptor differentiation. Importantly we observed a significant reduction in RPCs and cellular proliferation at the highest dose of the inhibitor, but reduced RGCs and photoreceptor specification at 2.5, 5 and 10 μM doses, suggesting that the latter two events may be more sensitive to TEAD inhibition. While activation of Hippo-Yap signalling has been studies in the context of Muller glia cell activation[74,75], our data provide important insights into the function into Hippo-Yap and related TEAD signalling during the very early stages of human retinogenesis.

The data generated herein have been submitted to open access online resources adding valuable information to the currently existing scRNA and ATAC-Seq to increase the sample and read size, but most importantly the use of ST to reveal the localisation of RPCs and neurogenic progenitors provides important insights into the long-debated source of retinal progenitors during human development. To assess if our data are molecularly distinct to those published recently, we compared our scRNA-Seq data to those published by Lu et al.[5], demonstrating an excellent correlation between the two data sets as exemplified by Fig. S11. Notably we would like to emphasise that unlike previous studies which have done either scRNA- or ATAC-seq on developing human retinas, or a combination of these

two methods on a small subset of samples up to 16 PCW, herein we report the successful application of both methods on a large sample dataset covering 7.5-21 PCW of human retinal development. Those combined with ST analyses, provide an integrated tempo-spatial single cell atlas of human retinal development up to midgestation.

## Methods

Human foetal material was provided by the Human Developmental Biology Resource (HDBR; www.hdbr.org). HDBR is licenced as a Research Tissue Bank by the UK Human Tissue Authority (licensing #: 12534) and operates in accordance with all applicable HTA Codes of Practice. The research complies with all relevant ethical regulations and was performed according to the protocols approved by the NHS Health Research Authority's Newcastle and North Tyneside 1 Research Ethics Committee (REC ref. 23:/NE/0135).

### Ethics

Embryo collection was performed following appropriate maternal written consent, and only after a decision to terminate a pregnancy had been made with a medical professional, in compliance with data protection governance as follows:

**Informed consent.** The consent procedure was overseen by a dedicated research nurse who has knowledge (i) of the issues surrounding the consent process; (ii) experience of recruiting participants to research projects and (iii) knowledge of the issues surrounding the care of women experiencing termination of pregnancy and miscarriage. The consenting procedure is briefly summarised below:

• Identification of suitable participants - This is performed in liaison with the clinical care team to ensure donors who are experiencing pain or are in too much of an emotional state for informed consent to be taken are not approached, following a clearly defined inclusion and exclusion criteria (this includes donors that do not use English as their first language, unless an interpreter is present; of if they are not deemed to be Gillick compliant).

• Provision of Study Information - A Participant Information Sheet (PIS) outlining what will happen should a donor choose to donate or not donate, a brief description of the type of research the material could be used for and who will use the donated material is provided to a potential donor. They are given at least 1 h to consider if they would like to donate, and given an opportunity to discuss any details with the research nurse to ensure they have all of the information they require to make an informed decision.

• Obtaining Informed Consent - Generic consent is given by the donor for their tissue to be used in ethically approved research projects. This is recorded using a written consent form (signed by the donor and the research midwife) and is filed in the donor¹s hospital records. The consent form ask specific questions, to which the donor must agree to all, in order to donate their tissue. These questions include asking if a donor has had the adequate opportunity to ask questions and has received satisfactory answers; and requesting permission for the analysis of genetic material of the donated tissue.

• Anonymous donation/Withdrawal of consent – The HDBR collects tissue anonymously, i.e. there is no link between the donor and the sample. Consequently, the donor is free to withdraw their consent at any time up to the point of tissue collection. Following this time, there is no possible way to link a sample back to a donor. However, donors are made aware via the PIS that should their personal details be held on other database that contain their genetic information (such as ancestry websites), there is a small but very unlikely possibility that this could be used to identify they have donated foetal tissue. The PIS also informs donors that they would not benefit from or be entitled to any profits resulting from the research, and that their treatment and care would not be affected should they choose to donate or not.

The participants were not compensated for sample donation.

**Data protection issues.** No personal data is held by the HDBR about the women who donate embryonic/foetal material for research. The material held in the HDBR is anonymous.

• Users of the HDBR agree that data will be made publicly available (after publication in journals, conferences etc), in a publicly accessible database available via the web. Data that will be available include the results from embryonic/foetal examination and from gene expression studies on embryonic/foetal cells, tissues and sections.

## scRNA- and -ATAC-Seq

11 samples of developing human eyes and 14 samples of neural retina (mixed gender) from 7.5–21 post-conception weeks (Supplementary Data 1 and 5) were obtained from the Human Developmental Biology Resource under ethics permission 23:/NE/0135 issued by the NorthEast Newcastle and North Tyneside 1 Research Ethics Committee. All samples were isolated and dissociated to single cells using a neurosphere dissociation kit (Miltenyi Biotech). Approximately 10,000 cells from each sample were captured, and sequencing libraries generated using the Chromium Single Cell 3′ Library & Gel Bead Kit (version 3, 10x Genomics). 10,000 of the subsequent nuclei were captured, and sequencing libraries generated using the Chromium Single Cell ATAC Library & Gel Bead Kit (version 1, 10x Genomics). Single cell RNA-Seq libraries were sequenced to 50,000 reads per cell and scATAC-Seq libraries were sequenced to 25,000 reads per nucleus on an Illumina NovaSeq 6000.

## scRNA-Seq analysis

The BCL files were de-multiplexed using CellRanger mkfastq version 3.01 and then aligned and quantify them against the human reference genome GRCh38 using Cellranger count. We performed quality control checks for each sample in R for each sample and removed any cells with <1000 reads or 500 genes or >10% mitochondrial reads. Any cells which expressed haemoglobin genes were also removed from the analysis. Doublets were predicted using DoubletFinder and filtered from the data. The sequencing depth was fairly consistent across samples. The complexity in the data increased most strongly with developmental ages rather than other technical features. There were some notable exceptions (e.g. 14556 retinal sample, Supplementary Data 1) which had a very low number of cells (326) after quality control and subsequently fewer cell types were detected. The Seurat R package (version 4.3.0) was then used to process the data prior to integration. Firstly, the raw data was normalised using the standard parameters. The FindVariableFeatures function used to select 2000 highly variable genes. The data was then scaled using ScaleData and the following variables were regressed out "percent.mt", "nCount_RNA", "nFeature_RNA". Principle component reduction with the 2000 highly variable genes selected, was applied to the scaled data using FindPCA function.

Harmony (version 0.1.1), which has been shown to preserve the smoothness in transitional populations[76], was used to remove sample batch effects from the data. A Uniform Manifold Approximation and Projection (UMAP) reduction was than applied to the first 10 harmony corrected components. The Seurat graph-based method was used to cluster the data. Resolutions from 0.2 to 2.2 were tested. Differential expression analysis using the standard settings in the FindMarkers function from Seurat were used to identify markers genes within each cluster. Cell types were then assigned to these clusters (Supplementary Data 1). Following this step we identified that clusters 36,40,41,42 at resolution 2.2 consisted of non-retinal cells (fibroblast/lens) and these were subsequently removed from the analysis. We then performed pseudotime on 4 branches of the UMAP, namely: RPC-T1-T2-T3, RPC, RPC-T1-RGC-T2-HC-AC, RPC-T1-T2-HC-AC, and RPC-T1-T3-Cones-Rods-BC. The data was subset by these cell type groups we re-clustered the data using the method described in the previous section. The cell types were re-annotated using retinal cell type specific marker genes (Supplementary Data 10) taken from published data, to ensure robust assignment of the cell types.

Monocle 3 was used to the order the cells and infer a pseudotime trajectory within the separate branches. We used Seurat FindAllMarkers to identify differentially expressed genes in the different cell types within each branch. The top 10 genes for each cell type, with the cells ordered by pseudotime order, were visualised in heatmaps.

## Comparison with published scRNA-Seq data

The count data from Lu et al.[5] was downloaded from GEO accessions GSE116106. Data was then loaded into Seurat (version 4.4.0) and filtered for cells with >2000 counts and 1000 features with <10% mitochondrial reads. Haemoglobin genes were removed prior to integration with our dataset. Doublets were removed using DoubletFinder 2.0.3. Integration was performed using SCTransform, with nCounts, nFeatures and percent.mt were passed to the vars.to.regress option. Batch correction was performed using harmony (version 1.1.0) with sample name and dataset passed to the group.by.vars option. A UMAP of the integrated datasets was then generated based on the harmony reduction, and plotted using a clustering resolution of 0.6. Cell types were annotated using marker genes as described in *scRNA-Seq analysis* section.

Pseudobulk samples were generated for each cell type from each dataset using Seurat's AggregateExpression function. Pseudobulk counts were then imported into DESeq2 (version 1.40.2), normalised using the vst function in DESeq2 and then batch corrected using limma (version 3.56.2). Principal components 1 and 2 were then plotted using DESeq2's plotPCA function.

## scATAC-Seq analysis

Peaks were detected using Cellranger ATAC software (version 1.2) in each of the samples. A set of shared peaks was then defined using Bedtools merge (version 2.30) and the Cellranger ATAC reanalyse function was then used to call peaks using the shared peak set. The datasets were imported using Signac and quality control steps were performed to remove cells low quality cells. We excluded cells with fewer than 20% of reads in peak region fragments, or <3000 peak region fragments. Cells with a TSS enrichment score of <2 and Blacklist ratio >0.05 or a nucleosome signal of <4 were also removed from downstream analysis.

Signac was then used to perform term frequency-inverse document frequency (TF-IDF) normalisation and singular value decomposition (SVD) dimension reduction for each individual sample. This was followed by UMAP reduction using components 2–30, and cluster analysis using Seurat. Signac was then used to generate a gene activity matrix based on open regions for each cell and FindAllMarkers from Seurat was used to predict upregulated genes for each cluster. These gene lists were used to assign cell type identity to the clusters. Retinal cell types were selected from each sample for integration and the normalisation, dimension reduction, and clustering steps, and cell type annotation steps described for the individual samples were applied to the combined dataset. We then identified differentially accessible peaks for each of the annotated cell types using the logistic regression (LR) test from the FindAllMarkers function. The average peak value for each cell type was calculated and differentially accessible peaks were plotted using the ComplexHeatmap package.

Chromvar was used to compute per cell motif activity scores for each cell and FindAllMarkers was used to compute enriched motifs for each cell type. The top enriched motifs ordered by average difference in z-score between each cell type are shown in a heatmap generated with the ComplexHeatmap package. Motif plots and Footprint plots were generated using Signac.

## Network analysis of scRNA-Seq and scATAC-Seq data

Using the Qiagen Ingenuity Pathway Analysis (IPA) software, we conducted a comprehensive analysis of the differentially expressed gene lists identified in RNAseq through IPA core analysis. IPA integrates manually curated knowledge from diverse biological data sources, encompassing gene expression, molecular interactions, and pathway information from an extensive experimental database. This integration allows for the identification of significant biological pathways. To perform the analysis the following steps were performed:

1. Load Data scRNAseq: The results of the differential expression analysis (Supplementary Data 2) were filtered to include p_val_adjust of 0.05 or less for each cell type and then uploaded to IPA. The gene symbols were set as ID with the "Gene symbol" option selected. The avg_log2FC and p_val_adj were used as the "Observations" with the options "Exprs Fold Change" and "Exprs False Discovery Rate" set respectively.
2. Run Core Analysis: A core analysis was run choosing the options "Expression Analysis" with the "Exprs Fold Change" selected to calculate the z-score. The analysis was run using "Ingenuity knowledge base (genes only). "Direct and indirect" relationships were considered. "Casual networks" were selected.
3. Network analysis: After running the core analysis we selected the "Upstream regulators" option then ordered the results by z-score to pick the networks with highest activation scores.
4. Comparison with ATACseq data: The differential accessibility data (Supplementary Data 6) split by cell type with p_val_adjust of 0.05 or less were loaded into IPA. The nearest gene nearest to the peak as determined by Cellranger ATAC was used in the ID column with the "Gene symbol" selected. The IPA overlay function was used to compare the observed accessibility profiles with the predicted regulators.

To further explore gene regulatory networks, we studied enhancer and gene regulatory interactions within our scRNA-Seq and scATAC-Seq data using SCENIC+ (version 1.0.1). pycisTopic (version 1.0.3) to identify and binarize cis-regulatory topics from our integrated ATAC dataset. Following this, pycistarget (version 1.0.3) was utilized to identify enriched motifs. We exported the integrated scRNA data into Python using scanpy (version 1.9.5) and created a joint RNA/ATAC object using SCENICplus, with the setting "multi_ome_mode = False." The "run_scenicplus" wrapper was executed to infer enhancer-driven gene regulatory networks (eGRNs). Regulon specificity scores (RSS) were calculated for each cell type, and these scores were presented as a heatmap to illustrate eGRN specificity for each cell type. We also generated plots for selected eGRNs.

## ST

Fresh frozen sections of four foetal retina samples of 8, 10, 11 and 13PCW (mixed gender, Supplementary Data 3) were used for the spatial transcriptome analyses performed with the Visium Spatial Gene Expression kit from 10XGenomics. First the tissue optimisation was performed defining 30 min as the most optimal permeabilization time window. The ST procedure was performed according to manufacturer's instructions. Four tissue sections from each sample were carefully placed into the four capture areas, fixed, haematoxylin and eosin stained and images in order to preserve histological information. This makes it possible to overlay the cell tissue image and the gene expression data in a later step. After permeabilization, reverse transcription reagents were added on top of the tissues. The tissues were subsequently removed, leaving the cDNA coupled to the arrayed oligonucleotides on the slide. The cDNA-RNA hybrids were cleaved off the chip and the sequencing libraries were prepared. The sequencing depth varied between 25,000,000 and 200,000,000 million reads.

## ST analysis

Spaceranger version 1.0 was used to demulitplex the data, generate gene expression matrices by aligning the FASTQs to GRCH38. The pipeline was also used to calculate spot co-ordinates using fiducial detection and to identify the spots covered by the tissue. The results from Spaceranger were imported into R using Spaniel (version 1.12). The data from each sample, which was derived from 4 consecutive sections within the tissue, was normalized and clustered, with a resolution of 0.5, using Seurat as described in the scRNA analysis section. FindAllMarkers was used to identify differentially expressed genes clusters. Spaniel was used to visualize the clusters overlaid on the tissue and generate spatial expression plots. The Seurat function "AddModuleScore" to the calculate an aggregate expression score of the early and late RPCs, and the T1, T2 and T3 neurogenic progenitors. The marker genes used for each cell type can be found at Supplementary Data 4.

## RNAscope

RNAscope in situ hybridization assay was used to determine the expression profile of *ZIC1, TFPI2, OPTC* and *HES6* during the development of the human retina. Formalin-fixed, paraffin embedded staged human foetal eyes were provided by the MRC/Wellcome Trust funded Human Developmental Biology Resource; (www.hdbr.org). Tissue sections were taken on a microtome at 8μm intervals to SuperFrost microscope slides and baked for 1 h at 60 °C before the paraffin was removed in xylene. The sections were first dehydrated in two changes of 100% ethanol before a target retrieval was performed by heating the sections for 8 min at 95 °C and incubating with a protease enzyme cocktail (ACD- Cat. No. 322381) for 15 min at 40 °C. RNAscope probes Hs-TFPI2 (ACD- Cat No. 470361), Hs-OPTC-C3 (ACD- Cat No. 1165211-C3), Hs-ZIC1-C2 (ACD- Cat. No. 542991-C2), Hs-PAX2 (ACD- Cat. No. 442541), Hs-HES6-C3 (ACD- Cat. No. 521301-C3) were hybridised to the tissue for 2 h at 40 °C followed by multiple rounds of signal amplification. Positive (ACD- Cat. No. 320861) and negative (ACD- Cat. No. 320871) control probes were used to confirm specificity. The annealed probes were detected using Opal fluorophores OPAL 570 (c1) OPAL 650 (C2) and OPAL 520 (C3) and imaged using a Zeiss microscope and ZEN software.

## Immunofluorescence analyses (IF)

Human foetal retinal tissue was fixed and IF performed on cryostat sections as previously described[2]. Sections were reacted against the following primary antibodies: CRX (1:200, Abnova, H00001406-MO2, Lot KB191-4G11), Ki67 (1:200, Abcam, ab15580, Lot GR3375617-1), Recoverin (1:1000, Millipore-Merck, ab5585, Lot 3099956), RXRγ (1:200, Santa Cruz Biotechnology, sc-555, Lot D1013), RXRγ (1:100, Santa Cruz Biotechnology,sc-365252, Lot G1122), SNCG (1:500, Antibodies.com, A121664, Lot 25309) and VSX2 (1:100, Santa Cruz, sc-365519, Lot 000031213). Secondary antibodies were conjugated to Alexa488 (Jackson Immuno Research Laboratories, 115-545-146, Lot 157251 and Thermo Fisher, 21202, Lot 2266877), Alexa546 (Thermo Fisher, 10040, Lot 1622582), Cy3 (Jackson Immuno Research Laboratories, 115-165-003, Lot 86185) and Alexa 647 (Thermo Fisher, 21447, Lot 2175459). All secondary antibodies were used 1:1000 diluted in PBS. Antibody specificity was assessed by omitting the primary antibodies. Images were obtained using a Zeiss Axio Imager.Z1 microscope with ApoTome.2 accessory equipment and AxioVision or Zen software. Between 5 and 10 images were collected from each IF analyses. Images are displayed as a maximum projection and adjusted for brightness and contrast in Adobe Photoshop CS6 (Adobe Systems).

## Retinal organoid differentiation

WT2 hiPSCs derived and characterised in our lab[77] were expanded in mTESR™1 (StemCell Technologies, 05850) on growth factor reduced Matrigel (BD Biosciences, San Jose, CA) coated plates at 37 °C and 5%

$CO_2$. Retinal organoids were generated as followed: hiPSCs were dissociated into single cells using Accutase (Gibco, A1110501) and seeded at a density of 7,000 cells/well onto U-bottom 96-well plates (Helena, 92697 T) prior coated with Lipidure (AMSbio, AMS.52000011GB1G) in mTeSR™1 with 10 μM Y-27632 ROCK inhibitor (Chemdea). After 2 days, 200 μl of differentiation medium as described in Dorgau et al.[78]. was added. Half of the differentiation medium was changed every 2 days until day 18 of differentiation. Then, the media was supplemented with 10% Foetal Calf Serum (FCS; Life Technologies, UK), Taurine (Sigma-Aldrich) and T3 (Sigma-Aldrich) and retinal organoids were transferred to 6-well low attachment plates (Corning, 3471). Retinoic Acid (RA; 0.5 μM; Sigma-Aldrich) was added from day 90 day 120 of differentiation. Media was changed every 2–3 days. MGH-CP1 (Sigma-Aldrich, T9032-10MG) at different concentrations (2.5 μM, 5 μM and 10 μM) or the vehicle control (DMSO; Sigma-Aldrich, D2650-5X10ML) were added to the culture media and refreshed at every media change. Retinal organoids were incubated with MGH-CP1 or DMSO for 2 weeks, starting at day 25 of differentiation and were collected after the incubation period at day 39 of differentiation.

## Statistics and reproducibility

The statistical methods for the single cell analyses are described in the appropriate sections above. For the rest of statistical analyses, Prism (GraphPad, USA) was used. Data were tested for normality using the D'Agostino & Pearson test. Comparisons between variables and statistical significance between groups were performed using One-way ANOVA or Kruskal-Wallis as nonparametric test followed by Dunnett's multiple comparisons test. No data were excluded from the analyses. Data were plotted as median values and percentiles unless indicated otherwise. Additional information such as $n$ values (the number of independent biological replicates) is presented in figure legends. Statistical significance of pair-wise comparisons is indicated by asterisks: $*p < 0.05$, $**p < 0.01$, $***p < 0.001$, and $****p < 0.0001$. Significance was defined as a $p$-value $< 0.05$.

## Reporting summary

Further information on research design is available in the Nature Portfolio Reporting Summary linked to this article.

## Data availability

All the single cell data generated in this study have been deposited to GEO under the accession numbers: GSE234971. Source data are provided with this paper.

## Code availability

The analyses scripts are described in: https://github.com/RachelQueen1/BBSRC_Retina[79]. The code has been archived on Zenodo with the [https://doi.org/10.5281/zenodo.10599556]. Request for bioinformatic pipelines and data analyses should be sent to rachel.queen@ncl.ac.uk.

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

## Acknowledgements

The authors are grateful for funding received from BBSRC UK (BB/T004460/1) to ML, RQ and JC and MRC UK (MR/S035826/1, MR/S036237/1) to ML, JCS and DRF. The human embryonic and foetal material was provided by the Joint MRC/Wellcome (MR/R006237/1, MR/X008304/1 and 226202/Z/22/Z) Human Developmental Biology Resource (https://www.hdbr.org/). The authors would like to acknowledge the Bioimaging Unit (Newcastle University), especially Dr Veronika Boczonadi, for their support and assistance in this work, and Prof. Roy Quinlan (University of Durham) for help with the definition of lens specific clusters in the ST analyses.

## Author contributions

B.D.—performed experiments, data collection and analyses, figure preparation. J.C.—performed experiments, data collection, fund raising. A.R., D.Z., M.C., R.H., J.C., T.D.—performed experiments, data collection. A.P., J.S., D.R.F.—provided key reagents, data discussion and fund raising. A.U.—bioinformatic data analyses. R.Q.—bioinformatic data analyses, data submission, figure preparation, contributed to manuscript writing and fund raising. M.L.—experimental design, data analyses, figure preparation, manuscript writing and fund raising.

## Competing interests

The authors declare no competing interests.
