## [Peer Review File · Nature Communications]

Single-cell analyses reveal transient retinal progenitor cells in the ciliary margin of developing human retinaREVIEWER COMMENTS

Reviewer #1 (Remarks to the Author):

In this manuscript the authors use scRNA, scATAC and Visium spatial transcriptomics to investigate the location and regulatory landscape of retinal progenitor cells. The authors generated 24 scRNA datasets from human embryos PCW 7.5-21 eyes and retinas and 12 scATAC-seq datasets from PCW 8.5-21 eyes and retinas, and 4 Visium spatial transcriptomics datasets from 8-13 PCW. Following identification of known cell types in their scRNA dataset, the authors performed pseudotime analysis to determine trajectories of progenitor cells. Next, they used the spatial transcriptomics data to investigate the layout of structures and celltypes in eyes from PCW 8.5 to 13 with a particular focus on the location of early and late retinal progenitor cells (RPCs). These datasets are utilized to demonstrate for the first time that early retinal progenitor cells present in the ciliary margin zone of humans between 6.6 and 10 PCW, consistent with observations in other organisms. Gene regulatory networks that drive these transitions are inferred from scATAC-seq data. Finally, the authors identified TEADs as potential regulator of RPC genes using motif analysis. To investigate the role of TEADs in the retina they treated retinal organoids with the inhibitor MGH-CP1 and determined that this reduced the number of RPCs and disorganized the retina.

The ideas of spatial gradient of retinal development described in the manuscript and the datasets are likely to be of broad interest. However, for reasons described below, the manuscript in its current form is not suitable for publication in Nature Communications: The methods description lacks sufficient detail to critically evaluate many findings; data in many instances should have been illustrated better for clearer interpretation; and some of the approach choices need justification.

Major comments

1. Lack of clarity in description of methods

In general, the manuscript lacks sufficient detail in the Methods to critically assess the findings presented. A detailed methods section with necessary detail on parameter settings are vital. Specific examples are listed below:

a. Cell-types: Identification of cell-types from single-cell data is a critical first step in any interpretation. No detail is provided on how cell-types were determined - was the analysis performed on a timepoint-by-timepoint basis or on the integrated data? A number of markers from previous single-cell studies are used - these should be compiled in a supplementary table with the correct references and representative markers illustrated on UMAPs.

b. Details on how the gene regulatory networks were derived is also not sufficiently described. The details on how Qiagen IPA tool works is missing and unfortunately the reader has no way to evaluate this approach. The authors should consider widely used gene regulatory network approaches such as SCENIC as an open source, publicly available alternative

c. Clustering of Visium data: It is not clear whether or not spatial information is used for clustering since the authors describe that the procedure used for clustering of spatial data was same as the single-cell data. Therefore, it is hard to assess what the claimed "spatially organized cell clusters" mean since this implies that the spatial layout of the clusters was involved in the clustering.

d. Gene expression signatures: The authors' analysis of the spatial data relies heavily on cell type expression signatures and the score is used in several plots as justification for the presence of RPCs in the CMZ in the PCW 8 spatial data, a key point in the paper. The exact definition of these scores must be in the methods. Further, the authors describe their derivation of the cell type signatures as follows: "we generated a gene expression signature for early and late RPCs from our pseudo-temporal analysis." It is unclear what the link is between pseudotime (the ordering of cells) and a gene expression signature for a cell type. It appears that markers of each celltype were identified and plotted over pseudotime, which is very distinct from using pseudotime to define markers.

2. Data quality control and data presentation

a. As acknowledged by the authors, the spot size of Visium which encompasses multiple cells poses a major challenge when interpreting the data. As a result, it is particularly important that the authors show gene expression of the marker genes on the spatial data. Plotting expression of marker genes on the spatial representation of the data itself would allow the reader to evaluate the strengths and weaknesses of this datatype and is very important for interpretation.

b. Related to 1a, the authors should demonstrate the expression of representative markers of genes on combined UMAPs. It is very hard for a reader to parse through large lists of differential expression without having representative examples on UMAPs. Differential expression by definition does not guarantee cluster-specific expression and hence in itself is not sufficient for the interpretation.

c. The authors do not show quality control metrics for their datasets. While it is great that UMAPs of all timepoints are presented in Fig S1, no information is provided about their read depths and molecule counts. This is very important as some of the datasets seem to have UMAPs with less structure and it would be helpful to know if this is due to a low number of cells or low read depth or representative of biology. Additionally, a strength of this paper is that it could be a good source of data for future analysis and therefore a detailed discussion about the quality of the data would be very beneficial.

3. Potential issues with analysis

a. Data integration with Harmony: Harmony makes an explicit assumption that the differences between datasets being integrated is in large part due to technical effects. The datasets generated by the authors are in a time course of development where there are a lot of important changes in cell abundances, types, and states. As such the differences between datasets are not always technical and hence a clear assessment of whether this batch correction approach is appropriate is necessary. This could be qualitative approaches like (1) Coloring the integrated UMAP by measured time points, (2) Plotting expression of markers of cell-types and terminal states, (3) Consistency of clustering between individual and integrated data types. Examples that indicate issues with integration: (A) the beginning of the emergence of “late born” cell types around PCW10 and the corresponding diminishing (as a proportion of the cells in the dataset) of “early born” types like RGCs starting at around that same time. (B) Figure 1A shows that RGCs (an early cell type) and amacrine cells (a late cell type) are pulled together in the UMAP despite being distinct cell types with apparently distinct transitions in datasets where they both appear with reasonably high abundance (PCW10, PCW12). (C) A similar issue may exist for rods and cones, though as those are both photoreceptors so that similarity may be more representative of the biology.

b. Trajectory detection using Monocle: The authors state “These analyses show that RGCs go through a T1 transition , horizontal and amacrine cells through a T1-T2 (Figure S2A) and photoreceptor and bipolar cells through a T1-T3 transition state (Figure S2B) ...”. This analysis involves multiple subsets of data separately analyzed for pseudotime. By subsetting the data the authors are placing their own biases into the analysis. It is possible that cells may transition through some of the excluded states to the terminal cell type of interest, this could change the pseudotime/trajectory analysis. Rather than subsetting the data, the authors should visualize the Monocle “principal graph”, the graph used to construct pseudotime, this graph allows the reader to see all the branches/trajectories in the data. Subset of branches can then be used for inferring transcriptional and regulatory dynamics rather than first subsetting the data.

c. The authors do not address motif redundancy and overlap. For example, the authors note that the motif PITX1 was enriched in photoreceptors and T3 and note that this is a novel finding. PITX1 has a binding motif that is extremely similar to OTX2. In the retina OTX2 plays a key role in defining the T3/photoreceptor/bipolar branch and appears to be a powerful chromatin remodeler. More information is needed to determine if the PITX1 motif is appearing in this analysis due to the activity of OTX2, or if PITX1 may play its own role. The authors should consider the use of Insilico-ChIP (<https://www.biorxiv.org/content/10.1101/2022.06.15.496239v1>), which takes into consideration not only the motif accessibility but also TF expression to derive more interpretable results.

4. Other comments

a. The authors state “To fully define the transitions from RPCs to T1, T2 and T3 neurogenic progenitors, we performed pseudo-temporal analysis of gene expression changes following cell cycle regression (Figure 1B). This analysis revealed bimodal densities of RPCs, mirroring the transition between early and

late RPCs described by Lu and colleagues 5, followed by the transitional T1 and then T2 and T3 neurogenic progenitors (Figure 1C, D).” No evidence is supplied to back up the claim of bimodal density.

b. The authors claim that with the inhibition of TEADs they get “inhibition of photoreceptor and retinal ganglion cell differentiation” in the treated organoids; however, they also note that they see a decrease in RPCs and in cycling cells. This implies that the RPCs are dying and/or no longer cycling (and thus being depleted by differentiating without renewal) in response to TEAD inhibition, and thus decreasing the number of neuronal cells. This is in line with the expected function of the TEAD family. This contrasts with the presented interpretation inhibition of cell-type specific differentiation where some lineages are being selectively targeted.

c. The authors state “The cone and rod precursors were characterized by high expression of THRB, and NRL and NR2E3 respectively, but interestingly also shared high expression of RGC and horizontal and amacrine cell markers (Figure S2B, Table S2).”. The expression of these genes should be plotted in some form in the supplement along with the staining.

Reviewer #2 (Remarks to the Author):

This manuscript titled “Spatiotemporal single cell analyses reveal a transient population of retinal progenitor cells in the ciliary margin of developing human retina” represents a remarkable amount of work performed across many different experimental modalities including single-cell (sc)RNA-Seq, scATAC-Seq, spatial transcriptomics, fluorescent RNA in situ hybridization, modeling of gene regulatory networks (GRNs) and pharmacologic treatment of retinal organoids. The reported single-cell results generally corroborate previous studies and nominate the TEAD family of transcription factors as a regulator of retinal progenitor cells. The spatial transcriptomic analysis suggests the presence of early retinal progenitors in the ciliary marginal zone (CMZ). The pharmacologic treatment of retinal organoids with the small molecule inhibitor of TEAD palmitoylation MGH-CP1 leads to a loss of marker expression of progenitor, photoreceptor, and ganglion and ganglion cells.

This manuscript is well-written and the technical aspects of this work, especially the single-cell analysis, appear to be well executed. These analyses will contribute depth to existing databases and could be used to determine reproducibility metrics. The inclusion of spatial transcriptomic analysis was a welcome novel aspect of this work. However, it is somewhat unclear how the finding that retinal progenitors transiently exist in the CMZ is different from the conventional understanding of the center-to-periphery pattern of retinal development in other vertebrates. Is the novel finding that this also occurs in humans? Given that this is reported as the major finding of the study, it was surprising that it was not investigated in greater depth. Showing that retinal organoid development is altered in a dose-dependent manner by treatment with MGH-CP1 is interesting, but as with the spatial transcriptomic studies, this direction would have benefited from more depth to support the intriguing, but seemingly preliminary

observations. Apart from this, the studies described here are commendable. Specific comments and questions include:

1) What is significance of the CMZ findings beyond what is already known in terms of the center-to-peripheral pattern of vertebrate retinal development? Isn't it expected that retinal progenitors with an "Early" molecular signature in the periphery would co-exist with RPCs with a "Late" molecular signature in the central retinal during development?

2) There is a population of cells in Figure 1B that seems to have been removed from panel 1A. What are these cells and why were they removed?

3) How are the scRNA-Seq and scATAC-Seq data or analyses distinct from what has been previously published (e.g. Lu et al., *Developmental Cell* 2020; Sridhar et al., *Cell Reports*, 2020, Finkbeiner et al., *Cell Reports*, 2021; Thomas et al., *Developmental Cell* 2022; Liang et al., *Cell Genomics*, 2023)? Given the challenges of working with human tissue, especially with respect to post-mortem interval and tissue integrity, how do the current datasets compare with the published studies in terms of cells and transcripts recovered? Can the new data be integrated successfully with previously published data or do they appear to be molecularly distinct in ways that can guide future human developmental studies?

4) No description of panel F in Figure 2 legend.

5) The description of the expression scales in the spatial transcriptomic panels Figure 2 & 3 are not sufficiently described to clearly understand what is being shown and the text is too small for easy interpretation. This makes it difficult to find the evidence for certain statements. For example, the authors write in the legend of Figure 3 "Note the scarce presence of early RPCs in the CMZ of the 13 PCW human eye", however to me it appears that there are more early RPCs in the CMZ of the human eye at 13 PCW compared to 10 or 11.

6) Figure 7 seems to be cut off or incompletely labeled.

7) Does treatment of developing organoids with MGH-CP1 cause them to de-differentiate or lose retinal identity altogether? This is a situation where the single-cell multiomic strategy could yield important novel insight.

8) Some of the most interesting and potentially controversial findings are reported without seeking to validate them through additional methods or to investigate them further. These reported findings include populations of proliferating cone photoreceptors and the computational prediction that Rhodopsin (presumably indirectly) regulates the expression of other phototransduction factors.

We would like to submit our revised manuscript titled: "Spatiotemporal single cell analyses reveal a transient population of retinal progenitor cells in the ciliary margin of developing human retina" to *Nature Communications* for further consideration. We thank the reviewers and the editorial board for providing very useful criticisms that have enhanced the value of our manuscript. We have considered the comments made by all reviewers. All of the new additions and corrections are highlighted in red throughout the main text. The reply to reviewers is show in blue font.

Reviewer #1

In this manuscript the authors use scRNA, scATAC and Visium spatial transcriptomics to investigate the location and regulatory landscape of retinal progenitor cells. The authors generated 24 scRNA datasets from human embryos PCW 7.5-21 eyes and retinas and 12 scATAC-seq datasets from PCW 8.5-21 eyes and retinas, and 4 Visium spatial transcriptomics datasets from 8-13 PCW. Following identification of known cell types in their scRNA dataset, the authors performed pseudotime analysis to determine trajectories of progenitor cells. Next, they used the spatial transcriptomics data to investigate the layout of structures and celltypes in eyes from PCW 8.5 to 13 with a particular focus on the location of early and late retinal progenitor cells (RPCs). These datasets are utilized to demonstrate for the first time that early retinal progenitor cells present in the ciliary margin zone of humans between 6.6 and 10 PCW, consistent with

observations in other organisms. Gene regulatory networks that drive these transitions are inferred from scATAC-seq data. Finally, the authors identified TEADs as potential regulator of RPC genes using motif analysis. To investigate the role of TEADs in the retina they treated retinal organoids with the inhibitor MGH-CP1 and determined that this reduced the number of RPCs and disorganized the retina.

The ideas of spatial gradient of retinal development described in the manuscript and the datasets are likely to be of broad interest. However, for reasons described below, the manuscript in its current form is not suitable for publication in Nature Communications: The methods description lacks sufficient detail to critically evaluate many findings; data in many instances should have been illustrated better for clearer interpretation; and some of the approach choices need justification.

Major comments:

1. Lack of clarity in description of methods

In general, the manuscript lacks sufficient detail in the Methods to critically assess the findings presented. A detailed methods section with necessary detail on parameter settings are vital. Specific examples are listed below:

a. Cell-types: Identification of cell-types from single-cell data is a critical first step in any interpretation. No detail is provided on how cell-types were determined - was the analysis performed on a timepoint-by-timepoint basis or on the integrated data? A number of markers from previous single-cell studies are used - these should be compiled in a supplementary table with the correct references and representative markers illustrated on UMAPs:

We thank the reviewer for this comment. The analysis was performed on a timepoint-by-timepoint basis as well as the integrated data. Instead of plotting each marker on a UMAP, we have preferred to present key marker information using dotplots. Dot plots showing the highly expressed markers per cell type is included with every sample in the revised Supplementary Tables S1, S3 and S5 and Figure 1B. The cell type definition was based on expression of retinal cell type specific markers previously reported. These are compiled into a Supplementary Table S10 now included with the revision.

b. Details on how the gene regulatory networks were derived is also not sufficiently described. The details on how Qiagen IPA tool works is missing and unfortunately the reader has no way to evaluate this approach. The authors should consider widely used gene regulatory network approaches such as SCENIC as an open source, publicly available alternative.

We agree that this a valuable addition to our manuscript. We have now extended our description of IPA, which makes use of a curated knowledge database (revised methods, page 19). We also have performed an additional analysis with SCENIC+ which makes inferences directly from the datasets themselves (Figure S7, revised text, pages 10, 11 and 20).

c. Clustering of Visium data: It is not clear whether or not spatial information is used for clustering since the authors describe that the procedure used for clustering of spatial data was same as the single-cell data. Therefore, it is hard to assess what the claimed "spatially organized cell clusters" mean since this implies that the spatial layout of the clusters was involved in the clustering.

We thank the reviewer for highlighting the unintentional ambiguity in phrasing. We did not use spatially informed clustering in this instance and have changed the wording accordingly (page 7). The change is from "revealing the presence of 12 spatially organised cell clusters" to "revealing the presence of 12 clusters relating to cell-types with distinct spatial locations".

d. Gene expression signatures: The authors' analysis of the spatial data relies heavily on cell type expression signatures and the score is used in several plots as justification for the presence of RPCs in the CMZ in the PCW 8 spatial data, a key point in the paper. The exact definition of these scores must be in the methods.

Spots within the clusters annotated as CMZ were selected. The Seurat function "AddModuleScore" was used to calculate an aggregate expression score of the Early RPC marker genes ("FGF19", "SFRP2", "DAPL1", "ZIC1", "ID3", "HMGA1", "EEF1A1", "TPT1", "TMSB4X", "MDK", "FOXP1", "GN2BL1", "HNRNPA1") for each spot within the CMZ region (revised methods, pages 20-21). The scores were then plotted as a violin plot. There was higher aggregate expression of early RPC marker genes in the CMZ in 8 PCW compared to 10PCW, 11PCW or 13PCW (Figure 3D), but no changes in aggregate

expression markers of late RPCs (Figure 3E). This has been added to revised results section (page 8).

Further, the authors describe their derivation of the cell type signatures as follows: "we generated a gene expression signature for early and late RPCs from our pseudo-temporal analysis." It is unclear what the link is between pseudotime (the ordering of cells) and a gene expression signature for a cell type. It appears that markers of each celltype were identified and plotted over pseudotime, which is very distinct from using pseudotime to define markers.

We thank the reviewer for this comment. To make it clearer in the text we have now revised the text to state " To investigate this further, we generated a gene expression signature for early and late RPCs, T1, T2 and T3 neurogenic progenitors based on the differentially expressed gene markers from the subsets of cells defined in our pseudo-temporal analysis (Figure 1F, Table S2). These gene expression signatures were further supplemented with marker genes described in published literature (Table S4) (page 7 of revised results section)".

2. Data quality control and data presentation

a. As acknowledged by the authors, the spot size of Visium which encompasses multiple cells poses a major challenge when interpreting the data. As a result, it is particularly important that the authors show gene expression of the marker genes on the spatial data. Plotting expression of maker genes on the spatial representation of the data itself would allow the reader to evaluate the strengths and weaknesses of this datatype and is very important for interpretation.

We have plotted the expression of key marker genes on the spatial plots of 8 PCW eye, this has now been included as Supplementary Figure S3C.

b. Related to 1a, the authors should demonstrate the expression of representative markers of genes on combined UMAPs. It is very hard for a reader to parse through large lists of differential expression without having representative examples on UMAPs. Differential expression by definition does not guarantee cluster-specific expression and hence in itself is not sufficient for the interpretation.

To reduce the number of figures/panels, while still showing the information requested by the reviewer, we have opted for a dotplot presentation for the integrated scRNA-Seq,

now included in Figure 1B. Dotplots are also generated for each stage of development both in the scRNA- and ATAC-Seq, and ST data in the revised Tables S1, S2 and S5.

c. The authors do not show quality control metrics for their datasets. While it is great that UMAPs of all timepoints are presented in Fig S1, no information is provided about their read depths and molecule counts. This is very important as some of the datasets seem to have UMAPs with less structure and it would be helpful to know if this is due to a low number of cells or low read depth or representative of biology. Additionally, a strength of this paper is that it could be a good source of data for future analysis and therefore a detailed discussion about the quality of the data would be very beneficial.

We have now added this information to the revised Tables S1, S2 and S5. The sequencing depth is fairly consistent across samples. The complexity in the data increases most strongly with developmental ages rather than other technical features. There are some notable exceptions (e.g. 14556 retinal sample) which had a very low number of cells (326) after QC and subsequently fewer cell types were detected. The methods (page 17) have been revised to reflect the very helpful suggestion made by the reviewer.

3. Potential issues with analysis

Data integration with Harmony: Harmony makes an explicit assumption that the differences between datasets being integrated is in large part due to technical effects. The datasets generated by the authors are in a time course of development where there are a lot of important changes in cell abundances, types, and states. As such the differences between datasets are not always technical and hence a clear assessment of whether this batch correction approach is appropriate is necessary.

This could be qualitative approaches like (1) Coloring the integrated UMAP by measured time points, Plotting expression of markers of cell-types and terminal states.

Consistency of clustering between individual and integrated data types. Examples that indicate issues with integration: (A) the beginning of the emergence of “late born” cell types around PCW10 and the corresponding diminishing (as a proportion of the cells in the dataset) of “early born” types like RGCs starting at around that same time.

We thank the reviewer for raising this point. We chose Harmony for batch correction because the method uses soft clustering which has been shown to preserve smoothness in transitional populations seen during development (PMID: 31740819). We have introduced this justification for software choice into our revised methods (page 18).

Furthermore, our experimental design consisted of multiple replicates for most developmental stages so we were able to adjust for technical effects between individual donors whilst preserving true biological differences seen between stages. Further the expression of cell type specific markers for each separate and integrated analysis has been added to revised Tables S1, S3 and S5 and Figure 1B.

To demonstrate the efficacy of our integration approach we are including the integrated UMAP showing the cell type annotations assigned during clustering of individual samples. This visualisation shows that the cell type clusters are maintained when integrating the samples.

Integrated scRNA-Seq data showing consistency of clustering between individual and integrated data types.

Integrated scATAC-Seq data showing consistency of clustering between individual and integrated data types.

Figure 1A shows that RGCs (an early cell type) and amacrine cells (a late cell type) are pulled together in the UMAP despite being distinct cell types with apparently distinct transitions in datasets where they both appear with reasonably high abundance (PCW10, PCW12).

We thank the reviewer for this comment; however we would like to point out that transcriptional studies performed in isolated rat amacrine and retinal ganglion cells show that the two cells display similar expression of most genes (> 74%) at both embryonic and postnatal stages in mice (PMID: 20445109). Thus it is not surprising that RGCs and amacrine cells are pulled together in the integrated UMAP shown in Figure 1A. The distinct state in 10 PCW and 12 PCW retinal UMAPS shown in **Figure S1**, reflects the earlier differentiation of RGCs (they are no longer being generated from RPCs/T1s), corroborating similar positions in day 82 fetal retina UMAP published by Sidhar and colleagues (PMID: 32023475).

A similar issue may exist for rods and cones, though as those are both photoreceptors so that similarity may be more representative of the biology.

Cones and rods are derived from the same transient neurogenic progenitor (T3) and located in the same retinal layer (ONL) and thus is not surprising that they are situated closely in the UMAP shown in Figure 1A. High correlation between expression levels of transcripts within rods and cones has already been reported by Lukowski et al. in the adult human retina (PMID: 31436334). Furthermore their side-by-side position in UMAPs has been shown in several recent publications of single cell RNA-Seq analyses in the fetal retina (PMID: 32023475, PMID: 32386599).

Trajectory detection using Monocle: The authors state "These analyses show that RGCs go through a T1 transition , horizontal and amacrine cells through a T1-T2 (Figure S2A) and photoreceptor and bipolar cells through a T1-T3 transition state (Figure S2B) ...". This analysis involves multiple subsets of data separately analyzed for pseudotime. By subsetting the data the authors are placing their own biases into the analysis. It is possible that cells may transition through some of the excluded states to the terminal cell type of interest, this could change the pseudotime/trajectory analysis. Rather than subsetting the data, the authors should visualize the Monocle "principal graph," the graph used to construct pseudotime, this graph allows the reader to see all the branches/trajectories in

the data. Subset of branches can then be used for inferring transcriptional and regulatory dynamics rather than first subsetting the data.

We appreciate the reviewer's comment. While we acknowledge that constructing a trajectory for all cell types can be effective in certain cases, we did not find this to be the case when analysing developmental retinal samples. Consequently, we opted to subset the data and study pseudotime following the approach outlined in the scRNA-seq development retinal analysis by Lu et al. 2020 (PMID: 32386599). We had two main reasons for this choice. *Firstly*, many cell types within the retina exhibit similar patterns of gene expression and we observed that trajectory analysis using all cell types missed crucial developmental branch points. However, by leveraging existing biological knowledge and evidence of retinal development to selectively choose specific cell types, we were able to explore different cell lineages and address highly specific questions about retinal development. *Secondly*, by adhering to the same approach as Lu et al. 2020 (PMID: 32386599), we could directly compare our findings with previous research and validate similarities in our analyses.

The authors do not address motif redundancy and overlap. For example, the authors note that the motif PITX1 was enriched in photoreceptors and T3 and note that this is a novel finding. PITX1 has a binding motif that is extremely similar to OTX2. In the retina OTX2 plays a key role in defining the T3/photoreceptor/bipolar branch and appears to be a powerful chromatin remodeler. More information is needed to determine if the PITX1 motif is appearing in this analysis due to the activity of OTX2, or if PITX1 may play its own role. The authors should consider the use of Insilico-ChIP (<https://www.biorxiv.org/content/10.1101/2022.06.15.496239v1>), which takes into consideration not only the motif accessibility but also TF expression to derive more interpretable results.

We thank the reviewer for this very useful comment. We did not find any networks with strong links to PITX1 using SCENIC and the expression of *PITX1* is only seen in a small percentage of cells which means that the method the reviewer suggested would not work. Hence, we have modified the text (revised results page 10) to state “Notably, shared binding motifs in T3 and photoreceptors were also highlighted for PITX1 and GSC TFs not associated previously with a role in rods or cones (Figure 5A), but those were not reproducible using SCENIC. Since PITX1 has a binding motif that is extremely similar to OTX2, a TF that plays a key role in defining the T3 differentiation to photoreceptor and

bipolar cells, it is likely that the results we have obtained are due to the similarity in the binding motifs between these two TFs”.

4. Other comments

a. The authors state “To fully define the transitions from RPCs to T1, T2 and T3 neurogenic progenitors, we performed pseudo-temporal analysis of gene expression changes following cell cycle regression (Figure 1B). This analysis revealed bimodal densities of RPCs, mirroring the transition between early and late RPCs described by Lu and colleagues 5, followed by the transitional T1 and then T2 and T3 neurogenic progenitors (Figure 1C, D).” No evidence is supplied to back up the claim of bimodal density.

Following the advice of the reviewer we have included a density plot of pseudotime scores within the RPC population (Figure 1D), which evidences the bimodal distribution corresponding to the early and late RPCs as described in the paper.

b. The authors claim that with the inhibition of TEADs they get “inhibition of photoreceptor and retinal ganglion cell differentiation” in the treated organoids; however, they also note that they see a decrease in RPCs and in cycling cells. This implies that the RPCs are dying and/or no longer cycling (and thus being depleted by differentiating without renewal) in response to TEAD inhibition, and thus decreasing the number of neuronal cells. This is in line with the expected function of the TEAD family. This contrasts with the presented interpretation inhibition of cell-type specific differentiation where some lineages are being selectively targeted.

We thank the reviewer for this comment. We have performed bulk RNA-Seq analysis of retinal organoids treated with 10 μ M MGH-CP1 and vehicle control. Our differential gene expression analyses (added at the end of results section, page 12 and Table S9) together with the quantitative immunofluorescence shows loss of retinal identity. Since the fraction of Caspase 3⁺ cells and VSX2⁺Casp3⁺ cells is very small, but the number of cycling VSX2⁺ RPCs is significantly reduced (Figure 7H and Figure S10C), we have concluded that loss of retinal identity is due to attenuation of RPC proliferation. The abstract and discussion (pages 13, 16) have been revised accordingly.

c. The authors state “The cone and rod precursors were characterized by high expression of THRB, and NRL and NR2E3 respectively, but interestingly also shared high expression

of RGC and horizontal and amacrine cell markers (Figure S2B, Table S2).". The expression of these genes should be plotted in some form in the supplement along with the staining. We thank the reviewer for this comment. To supplement the immunofluorescence staining we have now added overlay expression plots for RXRG and SNCG in the 10 PCW scRNA-Seq data (Figure S3A) showing expression of RGC marker SNCG in the RGC and cone photoreceptor clusters.

Reviewer #2 (Remarks to the Author):

This manuscript titled "Spatiotemporal single cell analyses reveal a transient population of retinal progenitor cells in the ciliary margin of developing human retina" represents a remarkable amount of work performed across many different experimental modalities including single-cell (sc)RNA-Seq, scATAC-Seq, spatial transcriptomics, fluorescent RNA in situ hybridization, modeling of gene regulatory networks (GRNs) and pharmacologic treatment of retinal organoids. The reported single-cell results generally corroborate previous studies and nominate the TEAD family of transcription factors as a regulator of retinal progenitor cells. The spatial transcriptomic analysis suggests the presence of early retinal progenitors in the ciliary marginal zone (CMZ). The pharmacologic treatment of retinal organoids with the small molecule inhibitor of TEAD palmitoylation MGH-CP1 leads to a loss of marker expression of progenitor, photoreceptor, and ganglion and ganglion cells.

This manuscript is well-written and the technical aspects of this work, especially the single-cell analysis, appear to be well executed. These analyses will contribute depth to existing databases and could be used to determine reproducibility metrics. The inclusion of spatial transcriptomic analysis was a welcome novel aspect of this work. However, it is somewhat unclear how the finding that retinal progenitors transiently exist in the CMZ is different from the conventional understanding of the center-to-periphery pattern of retinal development in other vertebrates. Is the novel finding that this also occurs in humans? Given that this is reported as the major finding of the study, it was surprising that it was not investigated in greater depth. Showing that retinal organoid development is altered in a dose-dependent manner by treatment with MGH-CP1 is interesting, but as with the spatial transcriptomic studies, this direction would have benefited from more

depth to support the intriguing, but seemingly preliminary observations. Apart from this, the studies described here are commendable. Specific comments and questions include:

1) What is significance of the CMZ findings beyond what is already known in terms of the center-to-peripheral pattern of vertebrate retinal development? Isn't it expected that retinal progenitors with an "Early" molecular signature in the periphery would co-exist with RPCs with a "Late" molecular signature in the central retina during development?

We thank the reviewer for raising this point. We have added a new paragraph at the discussion (pages 14, 15), highlighting the fact that early RPCs were also identified in the central retina of 8 and 10 PCW stages in addition to the CMZ (Figures 2H, 3A). This could suggest that two sources of RPCs could be found transiently in the developing human retina similar to prenatal mice. As for the co-existence of the early RPCs in the periphery with late RPCs in the central retina, this is exemplified in Figures 2H, J and Figure 3A-C. The relationship between CMZ and centrally located RPCs cannot be answered from our current data, but given the reports of CMZ-like presence in human retinal organoids, cell lineage tracing can be performed in this system through cell barcoding and single cell analyses. This has also been added to the revised discussion, pages 14-15.

2) There is a population of cells in Figure 1B that seems to have been removed from panel 1A. What are these cells and why were they removed?

We thank the reviewer for highlighting this discrepancy in our figure. Following integration, we removed a small number of non-retinal cells. We have now added the following sentence to our revised methods (page 17) "Following this step we identified that clusters 36, 40, 41, 42 at resolution 2.2 consisted of non-retinal cells (fibroblast/lens) and these were subsequently removed from the analysis" and revised Figure 1C.

3) How are the scRNA-Seq and scATAC-Seq data or analyses distinct from what has been previously published (e.g. Lu et al., Developmental Cell 2020; Sridhar et al., Cell Reports, 2020, Finkbeiner et al., Cell Reports, 2021; Thomas et al., Developmental Cell 2022; Liang et al., Cell Genomics, 2023)? Given the challenges of working with human tissue, especially with respect to post-mortem interval and tissue integrity, how do the current datasets compare with the published studies in terms of cells and transcripts recovered? Can the new data be integrated successfully with previously published data or do they

appear to be molecularly distinct in ways that can guide future human developmental studies?

We thank the reviewer for this very insightful comment. We have compared our scRNA-Seq data to those published by Lu et al. 2020 (PMID: 32386599), demonstrating an excellent correlation between the two data sets as exemplified by **Figure S11**. We attempted to integrate our scATAC-Seq data with two published datasets (PMID: 35081356, PMID: 35303433), however due to differences in peak calling the data integration was not possible. Given the excellent correlations observed with scRNA-Seq, we attribute the lack of integration of scATAC-Seq data to technical differences in methodology rather than molecular distinction between datasets. On this note, however we would like to emphasise that unlike previous studies which have done either scRNA- or ATAC-seq on developing human retinas, or a combination of these two methods on a small subset of samples up to 16 PCW, herein we report the successful application of both methods on a large sample dataset covering 7.5-21 PCW of human retinal development. Our dataset is the first retinal single cell dataset to be collected within the UK in alignment with the objectives of the Human Cell Atlas which requires a minimum of 20 samples to be collected from at least three geographically distinct sites. This combined with ST analyses, provides the first integrated atlas of human retinal development to date. This information has been added to the revised discussion, pages 16-17.

4) No description of panel F in Figure 2 legend.

We have added this to revised Figure 2 legend, page 29.

5) The description of the expression scales in the spatial transcriptomic panels Figure 2 & 3 are not sufficiently described to clearly understand what is being shown and the text is too small for easy interpretation. This makes it difficult to find the evidence for certain statements. For example, the authors write in the legend of Figure 3 “Note the scarce presence of early RPCs in the CMZ of 10-13 PCW human eye”, however to me it appears that there are more early RPCs in the CMZ of the human eye at 13 PCW compared to 10 or 11.

We have added new panels (Figure 3D, E) and text in both revised results (page 8) and methods (page 20) to substantiate the decreasing presence of early RPCs from 8PCW onwards.

6) Figure 7 seems to be cut off or incompletely labeled.

We thank the reviewer for this comment, but would like to highlight that the figure has not been cut off. The smaller bottom panel insets in C and E panel show individual marker and nuclear staining. This information has been added to revised figure legends, page 31.

7) Does treatment of developing organoids with MGH-CP1 cause them to de-differentiate or lose retinal identity altogether? This is a situation where the single-cell multiomic strategy could yield important novel insight.

We thank the reviewer for this comment. We have performed bulk RNA-Seq analysis of retinal organoids treated with 10 μ M MGH-CP1 and vehicle treated controls. Our differential gene expression analyses (added at the end of results section, page 12 and Table S9) together with the quantitative immunofluorescence shows loss of retinal identity. Since the fraction of Caspase 3⁺ cells and VSX2⁺Casp3⁺ cells is very small, but the percentage of cycling VSX2⁺ Ki67⁺ RPCs is significantly reduced, we have concluded that loss of retinal identity is due to attenuation of RPC proliferation. The abstract and discussion (page 16) have been revised accordingly.

8) Some of the most interesting and potentially controversial findings are reported without seeking to validate them through additional methods or to investigate them further. These reported findings include populations of proliferating cone photoreceptors.

We have validated the single cell RNA-Seq data by providing immunofluorescence staining showing co-expression of cone precursor (RXRG) with retinal ganglion cell (SNCG) marker in Figure S3B (page 6 of results section and Figure S3 legend of Supplementary Information). This analysis has shown that some of the RXRG-SNCG co-expressing cells in the outer nuclear layers were in a proliferative state at 8 PCW, but not in the later stages of development. We have corroborated these findings in human retinal organoids at day 45 of differentiation and have added this information in the revised results section (page 6).

and the computational prediction that Rhodopsin (presumably indirectly) regulates the expression of other phototransduction factors.

We thank the reviewer for this comment. We have searched through published data and have found several corroborating pieces of evidence which may indirectly link Rhodopsin to regulation of phototransduction genes as follows: 1) Human retinal organoids with additional transcriptionally active intact copies of Rhodopsin display increased expression of phototransduction genes (*PRPF*, *HCN1*, *SAG*; PMID: 36909455); 2) Human retinal organoids treated with Eupatilin which display a significant reduction in Rhodopsin expression (over 50%), also demonstrate reduced expression of essential phototransduction genes (*CRB1*, *EYES*, *SAG*, *GRK7*, *USH2A*, *PROM1*, *ARR3*, *PDE6A*, *GNGT1*; PMID: 37371046) and 3) Differential gene expression analysis of three week-old mouse model rhodopsin Q344X characterised by reduced expression of Rhodopsin, identified 12 differentially expressed genes belonging to the phototransduction pathway, compared to wild type mice (PMID: 29463953). A brief description of this corroborating evidence is added at the end of this result section, page 11 of revised manuscript.

We hope that these revisions are satisfactory. We look forward to hearing from you in due course.

REVIEWER COMMENTS

Reviewer #1 (Remarks to the Author):

NCOMMS-23-31180

I appreciate the authors taking time to thoroughly address the concerns raised in the initial review. The manuscript is substantially improved as a result of the revisions made the authors and for the most part, the analyses and their presentation are better accessible to the reader. The conclusions made by the authors are in-large part supported by the analyses but additional clarifications are necessary in my opinion:

1. Clarity of methods:

1. The description of methods using IPA software is still rather lacking. The main issue is the lack of documentation on the specific steps performed using this software without which it is difficult for a user to reproduce the analysis. The use of Scenic+ is a welcome addition however.

2. I appreciate the changes that the authors made to explain the aggregate marker scores. I suggest the following small updates to the figures to make it easier for the reader to tie the plots back to the new explanation:

- In Figures 2 G, H, I, J plots are still labeled as “early RPCs” and “late RPCs” this should be updated to something like “early RPC aggregate marker score” and “late RPC aggregate marker score”, to differentiate between cell counts (which is how they could currently be interpreted) and the scores. This information is in the figure legend, but it should also be on the plot as the current figure title is not descriptive and could easily be misinterpreted.

- Similar work should be done for Figure 3 (excluding Figure 3F). This figure is particularly hard to interpret as it is unclear if the early RPCs and late RPCs refer to cluster (as on the left side of Figure 3A-C), expression, or cell counts. Improving the labels of these plots will decrease confusion for future readers.

- On a related note the legend of Figure 2 reads “***G** and **I**”) Expression violin plots showing the **highest** aggregate expression scores for early RPCs in the peripheral retina (CMZ) and late RPCs in the central retina, respectively.” It is unclear what highest refers to in this context.

- Was differential expression performed on cells in the object from 1F, and then those markers were used were used for the signature? (in conjunction with markers from the literature.) As written mentioning pseudotemporal analysis makes me think that Monocle3::graph_test() or some other method of finding genes whose expression associated with pseudotime was used as opposed to doing DE on the object from Figure 1F. Further work on the wording would be appreciated.

2. Potential issues with analysis

1. Integration using Harmony. The authors say that soft clustering preserves structure even after integration and cite the Harmony paper. The paper does say that, but as you noted the assumption of differences being batch still holds. Specifically, rods and cones are found in separate clusters in **all** of the individual datasets in Figure S1, while in the integrated UMAP they have been thoroughly intermixed. A similar pattern largely holds for RGCs and Amacrine cells.

2. Monocle/Trajectory Subsetting:

The major problem here is that other transitions appear to have been removed, leaving only the transition of interest. For example, the authors state that the horizontal and amacrine cells go through a T2 transition. Inspection of Figure S2A reveals that the authors have subset the cells to the following types; Horizontals, Amacrines, T2, T1, and progenitors. The authors have removed the possibility that Horizontal and Amacrine cells could at any point in this transition occupy another state (such as T3). This is akin to removing all roads in a town but one, and then remarking that everyone is driving down it.

The problem here is not that I believe that the authors are incorrect in their statement that Horizontals and Amacrines travel through T1/T2, RGCs travel through T1, and photoreceptors and bipolars travel through T3, but the statement “These analyses show that RGCs go through a T1 transition (data not shown), horizontal and amacrine cells through a T1-T2 (Figure S2A) and photoreceptor and bipolar cells through a T1-T3 transition state (Figure S2B), corroborating recently published scRNA-Seq data on few stages of human fetal retinal development.” Implies that cells were presented other options and yet chose to move down these transitions, rather than that all other options were removed. In this case the authors have modified the data to make their hypothesis the only option without making it clear to the reader, and then later claiming that the data supports their hypothesis.

Further the authors state in their rebuttal that Monocle 3 did not represent these transitions well, I recommend experimenting with the `learn_graph_control` parameters for `Monocle3::learn_graph` (you can see how to use it here: https://rdrr.io/github/cole-trapnell-lab/monocle3/man/learn_graph.html). You can use these parameters to increase the flexibility of the graph and will sometimes make it better at finding a good path through transitions. Additionally other tools such as Palantir can be used to calculate branch probabilities and may provide better resolution.

1. I appreciate that the authors have updated the results to reflect the similarity between PITX and OTX2. The authors note in their rebuttal that Insilico-ChIP would not work because expression of PITX is low. This information should be shared with the reader as it implies that this gene may not be expressed in this context and is unlikely to be the source of his motif abundance. (Though obviously this does not eliminate this possibility completely as it could have been expressed in a precursor and the protein may remain.)

2. Minor typos:

- Accidental capitalization: “high expression of PROX1, a marker of retinal progenitors, horizontal and ****AII**** amacrine cells”
- Misspelling of the word the: “and additionally displayed ****ethe**** expression of typical photoreceptor precursors (OTX2, CRX) and bipolar cell markers (VSX1)”

Reviewer #2 (Remarks to the Author):

My questions and concerns have been addressed through the revisions made by the authors.

We thank the reviewers for providing very useful criticisms that have enhanced the value of our manuscript. We have considered the comments made by all reviewers. All of the new additions and corrections are highlighted in red throughout the main text. The reply to reviewers is shown in blue font.

Reviewer #1:

I appreciate the authors taking time to thoroughly address the concerns raised in the initial review. The manuscript is substantially improved as a result of the revisions made by the authors and for the most part, the analyses and their presentation are better accessible to the reader. The conclusions made by the authors are in large part supported by the analyses but additional clarifications are necessary in my opinion:

1. Clarity of methods:

1. The description of methods using IPA software is still rather lacking. The main issue is the lack of documentation on the specific steps performed using this software without which it is difficult for a user to reproduce the analysis. The use of Scenic+ is a welcome addition however.

We have provided a detailed IPA method description in the revised Methods section (page 20).

2. I appreciate the changes that the authors made to explain the aggregate marker scores. I suggest the following small updates to the figures to make it easier for the reader to tie the plots back to the new explanation:

- In Figures 2 G, H, I, J plots are still labeled as “early RPCs” and “late RPCs” this should be updated to something like “early RPC aggregate marker score” and “late RPC aggregate marker score”, to differentiate between cell counts (which is how they could currently be

interpreted) and the scores. This information is in the figure legend, but it should also be on the plot as the current figure title is not descriptive and could easily be misinterpreted.

The changes have been implemented, please see revised Figure 2 (panels G-J).

- Similar work should be done for Figure 3 (excluding Figure 3F). This figure is particularly hard to interpret as it is unclear if the early RPCs and late RPCs refer to cluster (as on the left side of Figure 3A-C), expression, or cell counts. Improving the labels of these plots will decrease confusion for future readers.

We thank the reviewer for this comment. We have now revised the Figures labelling the panels with “aggregate scores” and “gene expression signatures”. Each spot is likely to contain more than one cells, hence we looked for gene expression signatures to define regions where early and late RPCs may be located. The revised 3E presents gene expression signatures superimposed on the spatial eye images, whilst 3F, G present gene expression aggregate scores. Panels 3A-D present cluster information. The Figure legends has been changed accordingly (page 30).

- On a related note the legend of Figure 2 reads “**G** and **I**”) Expression violin plots showing the **highest** aggregate expression scores for early RPCs in the peripheral retina (CMZ) and late RPCs in the central retina, respectively.” It is unclear what highest refers to in this context.

This has been revised, please refer to Figure 2 legend (page 30).

- Was differential expression performed on cells in the object from 1F, and then those markers were used for the signature? (in conjunction with markers from the literature.) As written mentioning pseudotemporal analysis makes me think that Monocle3::graph_test() or some other method of finding genes whose expression associated with pseudotime was used as opposed to doing DE on the object from Figure 1F. Further work on the wording would be appreciated.

We did not use graph test, but did differential gene expression between cell types defined in our subset in PT analyses in 1F and then complemented this analysis with the published literature. The result section has been changed accordingly (page 7).

2. Potential issues with analysis

1. Integration using Harmony. The authors say that soft clustering preserves structure even after integration and cite the Harmony paper. The paper does say that, but as you noted the assumption of differences being batch still holds. Specifically, rods and cones are found in separate clusters in ****all**** of the individual datasets in Figure S1, while in the integrated UMAP they have been thoroughly intermixed. A similar pattern largely holds for RGCs and Anacrine cells.

We agree that the cones and rods are less distinct in the fully integrated data. However we display below the subsetted data showing individual cell type annotations superimposed. These plots show that cones and rods are distinct. This supports both our choice of integration and the reason we choose to subset the data to better study the separate lineages.

2. Monocle/Trajectory Subsetting:

The major problem here is that other transitions appear to have been removed, leaving only the transition of interest. For example, the authors state that the horizontal and amacrine cells go through a T2 transition. Inspection of Figure S2A reveals that the authors have subset the cells to the following types; Horizontals, Amacrine, T2, T1, and progenitors. The authors have removed the possibility that Horizontal and Amacrine cells could at any point in this transition occupy another state (such as T3). This is akin to removing all roads in a town but one, and then remarking that everyone is driving down it.

The problem here is not that I believe that the authors are incorrect in their statement that Horizontals and Amacrine travel through T1/T2, RGCs travel through T1, and

photoreceptors and bipolars travel through T3, but the statement “These analyses show that RGCs go through a T1 transition (data not shown), horizontal and amacrine cells through a T1-T2 (Figure S2A) and photoreceptor and bipolar cells through a T1-T3 transition state (Figure S2B), corroborating recently published scRNA-Seq data on few stages of human fetal retinal development.” Implies that cells were presented other options and yet chose to move down these transitions, rather than that all other options were removed. In this case the authors have modified the data to make their hypothesis the only option without making it clear to the reader, and then later claiming that the data supports their hypothesis.

Further the authors state in their rebuttal that Monocle 3 did not represent these transitions well, I recommend experimenting with the learn_graph_control parameters for Monocle3::learn_graph (you can see how to use it here: https://rdr.io/github/cole-trapnell-lab/monocle3/man/learn_graph.html). You can use these parameters to increase the flexibility of the graph and will sometimes make it better at finding a good path through transitions. Additionally other tools such as Palantir can be used to calculate branch probabilities and may provide better resolution.

We thank the reviewer for this very useful comment. To address his we have added a new supplementary Figure (S2C) in which the integrated single cell RNA-Seq data UMAP is plotted with the trajectory superimposed. This clearly shows that the horizontal and amacrine cells go through a T2 transition. We have also added this in the revised results section, page 6.

1. I appreciate that the authors have updated the results to reflect the similarity between PITX and OTX2. The authors note in their rebuttal that Insilico-ChIP would not work because expression of PITX is low. This information should be shared with the reader as it implies that this gene may not be expressed in this context and is unlikely to be the source of his motif abundance. (Though obviously this does not eliminate this possibility completely as it could have been expressed in a precursor and the protein may remain.)

We have added this information to the revised results section, page 9.

2. Minor typos:

- Accidental capitalization: “high expression of PROX1, a marker of retinal progenitors, horizontal and ****All**** amacrine cells”

We are unsure what the reviewer means by this comment, All represent a subset of amacrine cells and All nomenclature is widely used to represent them, hence we have not amended this sentence.

- Misspelling of the word the: “and additionally displayed **ethe** expression of typical photoreceptor precursors (OTX2, CRX) and bipolar cell markers (VSX1)”

This has been corrected, please refer to revised results section, page 6.

Reviewer #2 (Remarks to the Author):

My questions and concerns have been addressed through the revisions made by the authors.

We hope that these revisions are satisfactory. We look forward to hearing from you in due course.

REVIEWERS' COMMENTS

Reviewer #1 (Remarks to the Author):

The authors have satisfactorily addressed the concerns raised in the review and I support the publication of this manuscript in Nature Communications.